# Large-scale forest stand height mapping in the northeastern U.S. and China using L-band spaceborne repeat-pass InSAR and GEDI LiDAR data

Yanghai Yu[1], Yang Lei[1], Paul Siqueira[2], Xiaotong Liu[3], Denuo Gu[4], Anmin Fu[3], Yong Pang[4], Wenli Huang[5], Jiancheng Shi[1]

[1]National Space Science Center, Chinese Academy of Sciences, Beijing, 100190, China
[2]Department of Electrical and Computer Engineering, University of Massachusetts, Amherst, MA 01003-9284, USA
[3]Academy of Forest Inventory and Planning, State Forestry Administration, Beijing, 100714, China
[4]Institute of Forest Resource Information Techniques, Chinese Academy of Forestry, Beijing, 100091, China
[5]School of Resource and Environmental Science, Wuhan University, Wuhan, 430079, China

*Correspondence to*: Yang Lei (leiyang@nssc.ac.cn)

**Abstract.** This paper presents a global-to-local fusion approach combining spaceborne Synthetic Aperture Radar (SAR) Interferometry (InSAR) and LiDAR to create large-scale mosaics of forest stand height. The forest height estimates are derived based on a semi-empirical InSAR scattering model, which links the forest height to repeat-pass InSAR coherence magnitudes. The sparsely yet extensively distributed LiDAR samples provided by Global Ecosystem Dynamics Investigation (GEDI) mission enables the parameterization of signal model at a finer spatial scale. The proposed global-to-local fitting strategy allows for the efficient use of LiDAR samples to determine adaptive model at a regional scale, leading to improved forest height estimates by integrating InSAR-LiDAR under nearly concurrent acquisition conditions. This is supported by fusing the ALOS-2 and GEDI data at several representative forest sites. This approach is further applied to the open-access ALOS InSAR data to evaluate its large-scale mapping capabilities. To address temporal mismatch between the GEDI and ALOS acquisitions, disturbances such as deforestation are identified by integrating ALOS-2 backscatter products and GEDI data. A modified signal model is further developed to account for natural forest growth over temperate forest regions where the intact forest landscape along with forest height remains quite stable and only change slightly as trees grow. In the absence of detailed statistical data on forest growth, the modified signal model can be well approximated using the original model at regional scale via local fitting. To validate this, two forest height mosaic maps based on the open-access ALOS-1 data were generated for the entire northeastern regions of the U.S. and China with total area of 18 million and 152 million hectares, respectively. The validation of the forest height estimates demonstrates improved accuracy achieved by the proposed approach compared to the previous efforts i.e., reducing from a 4.4 m RMSE at a few-hectare pixel size to 3.8 m RMSE at a sub-hectare pixel size. This updated fusion approach not only fills in the sparse spatial sampling of individual GEDI footprints, but also improves the accuracy of forest height estimates by 20% compared to the interpolated GEDI maps. Extensive evaluation of forest height inversion against LVIS LiDAR data indicates an accuracy of 3-4 m over flat areas and 4-5 m over hilly areas in the New

England region, whereas the forest height estimates over the northeastern China are best compared with small footprint LiDAR validation data even at an accuracy of below 3.5 m and with a coefficient of determination (R2) mostly above 0.6. Given the achieved accuracy for forest height estimates, this fusion prototype offers a cost-effective solution for public users to obtain wall-to-wall forest height maps at a large scale using freely accessible spaceborne repeat-pass L-band InSAR (e.g., forthcoming NISAR) and spaceborne LiDAR (e.g., GEDI) data.

## 1 Introduction

Forests play a crucial role in the terrestrial ecosystem as they serve as one of the largest terrestrial carbon pool (Pan et al., 2011). As identified by IPCC v6 (Masson-Delmotte et al., 2021) and international forest monitoring efforts such as United Nation's REDD+ programme (Angelsen, 2009), large-scale (e.g., state, continental, and global) forest height products are desired to quantify carbon storage in forested resources due to their close relationship to aboveground biomass (AGB). These products also help to determine forests' roles in climate change mitigation and biodiversity conservation (Houghton et al., 2009). In this work, "forest height" or "forest stand height" (FSH) are referred to as the medium-footprint (25 m) LiDAR-determined relative height at the 98th percentile (rh98) as measured by NASA's GEDI LiDAR mission onboard the International Space Station (ISS).

Satellite-based remote sensing represents a cost-effective method to investigate biophysical parameters of forests. Commonly used remote sensing methods include optical and microwave imaging observations, such as passive optical sensors including Landsat series (Loveland and Dwyer, 2012), LiDAR missions including ICESat-1/2 (Schutz et al., 2005; Abdalati et al., 2010) and GEDI (Dubayah et al., 2020) missions, and SAR systems such as JAXA's ALOS-1/2 (Rosenqvist et al., 2007; Rosenqvist et al., 2014), Sentinel-1 (Torres et al., 2012), TanDEM-X (Krieger et al., 2007). LiDAR and SAR are promising for capturing the internal vertical structure of forests: LiDAR is fundamentally sensitive to structural details, while radar detects the three-dimensional distribution of vegetation elements (Ulaby et al., 1990). The backscatter information from a single SAR image can be used for inferring the AGB (Santoro et al., 2021), despite the fact that the actual vertical information remains undetermined. As an extension of SAR backscatter observation, SAR interferometry (InSAR) provides direct information related to the vertical forest structure (Treuhaft and Siqueira, 2000). Spaceborne InSAR can operate either by single-pass bistatic interferometer (e.g., TanDEM-X missions (Krieger et al., 2007)), or by repeat-pass InSAR (e.g., ALOS-1/2 L-band, Sentinel-1 C-band missions). The short wavelength operated in former satellites may restrict its sensing capabilities over dense forests, while temporal decorrelation affects repeat-pass InSAR performance (Zebker and Villasenor, 1992; Monti-Guarnieri et al., 2020; Lavalle et al., 2012; Ahmed et al., 2011). LiDAR has been widely used for characterizing the forest vertical structure at a regional scale, and can serve as a benchmark for calibrating inversion models and forest height estimates (Choi et al., 2023; Askne et al., 2013). Spaceborne LiDAR (Schutz et al., 2005) has been further developed for global ecosystem

monitoring. Because of observational constraints, these measurements have been acquired based on such a spatial sampling technique that collect sparse yet extensive measurements.

For instance, NASA's GEDI mission is the first spaceborne LiDAR instrument designed to study ecosystems. Since 2019, GEDI has provided extensively distributed LiDAR waveform measurements covering nearly all global forests. These waveform observations allow for the extraction of various biophysical parameters, such as canopy height and leaf area index. However, GEDI collects only discrete footprint measurements, spaced approximately 60 meters apart in the along-track direction and 600 meters apart in the cross-track direction. To overcome this limitation and extend GEDI's measurements into continuous datasets, several fusion studies have been conducted. Notable examples include efforts that incorporate radiometric information from optical sensors, such as NASA's Landsat (Potapov et al., 2021) and ESA's Sentinel-2 (Lang et al., 2022), as well as from SAR backscatter signals (Shendryk, 2022). However, relying solely on radiometric information to expand LiDAR observations has proven suboptimal, particularly in high-biomass regions where signal saturation occurs (Kalacska et al., 2007; Imhoff, 1995; Dinh Ho Tong Minh et al., 2014).

In contrast, because of its fundamental sensitivity to height and/or variations in height, the fusion of SAR interferometry and GEDI has gained much interests. For example, making joint use of TanDEM-X and GEDI data has been assessed and demonstrated for achieving wall-to-wall forest height and AGB mapping (Qi and Dubayah, 2016; Choi et al., 2023; Guliaev et al., 2021; Qi et al., 2019). Without temporal decorrelation effects, TanDEM-X data offer opportunities to leverage very-high-resolution observations for addressing spatially heterogeneous landscapes. However, the forest height was inverted in these studies based on the Random Volume Over Ground (RVoG) model (Treuhaft and Siqueira, 2000, Cloude and Papathanassiou, 1998) and an external constraint induced by GEDI waveform information. That is, only the mean waveform information across the scene in these studies was used for model-based inversion, implying an underlying assumption that the forest objects over the scene share a similar vertical structure. This may lead to a degraded performance when dealing with spatially heterogeneous forests. To address this, (Qi et al., 2025) proposed a regional post-processing correction model to refine suboptimal height estimates, while (Hu et al., 2024) exploited local ICESat-2 LiDAR information, using regional polynomials and an adaptive window, to estimate equivalent forest phase centers under homogeneous forest and terrain conditions. Additionally, a potential limitation of TanDEM-X observations is the insufficient penetration capability over dense forests due to the short wavelength of the X-band (~3.1 cm) (Kugler et al., 2014). This underlines the need for longer-wavelength SAR systems (e.g., L-band) to enhance sensitivity in dense forest environments.

Temporal decorrelation has been a widely studied topic in InSAR research (Rocca, 2007; Ahmed et al., 2011; Bhogapurapu et al., 2024). (Zebker and Villasenor, 1992) proposed a Gaussian model to analyze oceanic scenarios, while (Monti-Guarnieri et al., 2020) summarized the signal models tailored for vegetated scenarios. (Askne et al., 1997) introduced a coordinate-dependence of the vertical motion profile to analyse InSAR temporal decorrelation effects caused by wind. Building upon the

well-known RVoG model, several signal models have been developed to explicitly incorporate temporal decorrelation effects (Lavalle et al., 2012; Papathanassiou and Cloude, 2003; Lei et al., 2017a).

This study employs the RVoG-based temporal decorrelation model (Lei et al., 2017a) to invert forest height. The model-based inversion was first demonstrated using a relatively small airborne LiDAR reference strip to generate a forest height mosaic over a two-state region in the northeastern U.S. (Lei et al., 2018). Given the limited availability of LiDAR datasets at that time, scene-wide constant model parameters for the relationship between the InSAR temporal decorrelation and LiDAR observations

were assumed and the overlapping area between InSAR scenes were used to propagate the LiDAR information throughout the adjacent InSAR scenes. The advent of NASA's Global Ecosystem Dynamics Investigation (GEDI) mission has since enhanced this methodology by integrating local GEDI samples directly into the inversion framework (Lei and Siqueira, 2022; Yu et al., 2023). This integration enables spatially adaptive calibration of model parameters, overcoming prior limitations of constant parameter assumptions and improving inversion accuracy in heterogeneous forest landcovers.

This paper further removes the assumption of spatially constant temporal change model that were made in the previous efforts, and develops a new inversion approach based on a two-stage (global-to-local) inversion strategy. By efficiently leveraging regional GEDI samples, this approach calibrates a semi-empirical, semi-physical repeat-pass InSAR model at a finer spatial scale, substantially improving forest height inversion accuracy. The method assumes that the temporal decorrelation model

remains spatially invariant at the regional scale, while permitting variability in forest height observations within those regions. This approach is validated by fusing ALOS-2 InSAR and GEDI data acquired under nearly concurrent conditions. Furthermore, the approach is applied to the open-access ALOS InSAR data for evaluating its large-scale mapping capability. To address the temporal mismatch between the ALOS and GEDI acquisition, forest disturbance can be detected by fusing SAR backscatter and LiDAR data under nearly concurrent condition. Furthermore, a modified model is developed to account for the natural

growth of forests over temperate forest regions where the Intact Forest Landscape (IFL) exhibits slow changes in height. Without available forest growth data, the modified signal model can be well approximated using the original signal model at regional scale through local fitting. Two 30 m gridded forest height mosaics were generated for the northeastern regions of U.S. and China. Validation of the generated forest height mosaics against extensive airborne LiDAR observations demonstrate enhanced inversion accuracy at sub-hectare pixel size. The key contribution of this paper lies in the use local GEDI information

for Radar-LiDAR data fusion, enabling large-scale and efficient forest height mapping using open-access spaceborne data, such as GEDI and forthcoming NISAR (Siqueira et al., 2024; Kellogg et al., 2020) data.

This paper is structured as follows: Section 2 describes the study areas and remote sensing datasets; Section 3 details the proposed inversion framework; Section 4 validates forest height estimates across diverse forest sites in the northeastern U.S.

and China. Section 5 discusses the implications and limitations of the methodology, and Section 6 concludes this study.

## 2 Study area and datasets

### 2.1 Study area

This paper focuses on the northeastern regions of the U.S. and China. These regions contain transitional forests composed of both coniferous and broad-leaved species. As shown in Figure 1, the New England region in the northeastern U.S. (including the states of Maine, New Hampshire, Vermont, Massachusetts, Connecticut, and Rhode Island) is selected for the generation and validation of the large-scale mosaic of forest height (covering a total area of 18 million hectares) due to the availability of ample airborne LiDAR datasets. Forests in this area are primarily dominated by coniferous forests (Red Pine, Balsam Fir, Hemlock etc.) and northern hardwoods (Maple, Oak, Beech, etc.). These forests exhibit stable intact forest landscapes (IFL) and canopy heights, with changes driven primarily by natural growth rather than anthropogenic disturbance (Riofrío et al., 2023).

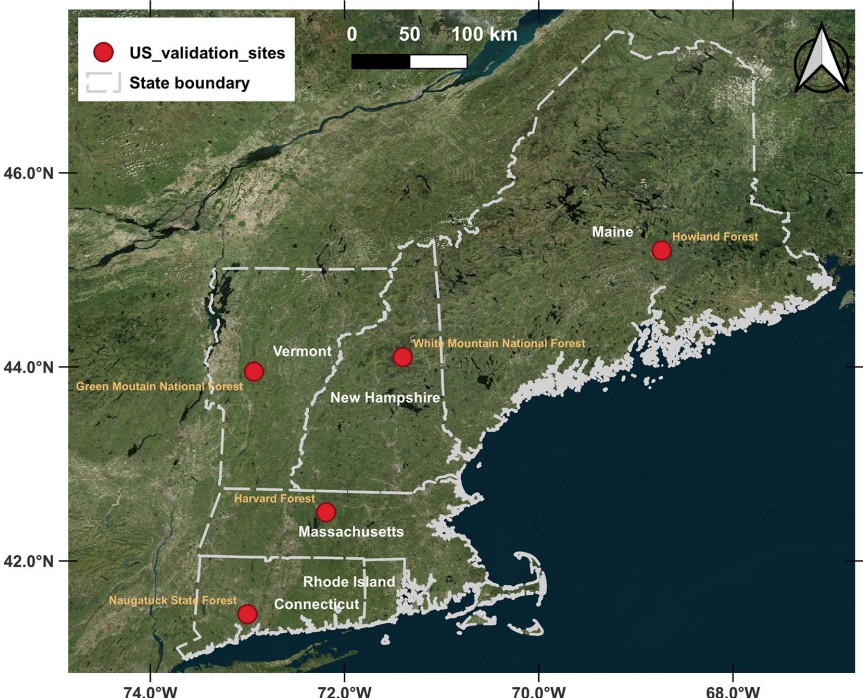

**Figure 1: Study area and validation sites for the New England region in the U.S. The generated forest height mosaic map covers the states of Maine, New Hampshire, Massachusetts, Vermont, Connecticut, and Rhode Island. The inversion results are validated against the large-footprint (25 m) LVIS data acquired either in 2009 or in 2021 over the validation sites (denoted by red dot markers). At the White Mountain National Forest (WMNF) site, small footprint GRANIT LiDAR data are also used for validation after reprocessing into equivalent RH98 metric maps. The features of the validation sites are summarized in Table 1.**

Another large-scale forest height mosaic is also generated over the northeast of China with a total area of 152 million hectares. As shown in Figure 2, the forest height mosaic for China covers five provinces: Hebei, Jilin, Liaoning, Inner Mongolia and

Heilongjiang. Note that Jilin and Inner Mongolia provinces were not fully covered as only forested areas within the GEDI observation coverage (<51.6° N) are addressed. The forests in northeastern China can be primarily grouped into four primary regions: 1) deciduous coniferous forest region located at the northernmost parts of Inner Mongolia and Heilongjiang provinces; 2) temperate mixed-forest region (comprising evergreen coniferous and deciduous broad-leaved species) primarily distributed in Heilongjiang and Jilin provinces; 3) northern temperate mixed-forest subregion situated in Liaoning province; and 4) temperate steppe region located partly in Hebei province and partly in Inner Mongolia province.

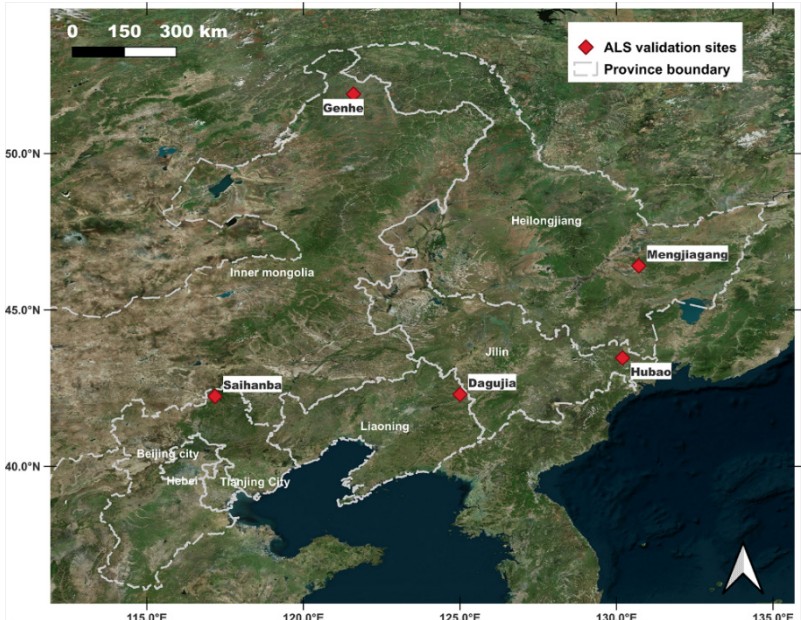

**Figure 2: Study area and validation sites in northeastern China. Five provinces are covered in the generated forest height mosaic: Jilin, Liaoning, Hebei, Heilongjiang, and Inner Mongolia. The performance of the forest height inversion is assessed by comparison with small-footprint (0.5-1 m) LiDAR data at the validation forest sites (indicated by the red diamond markers) in each province. The features of the validation sites are summarized in Table 2.**

The northeastern regions of the U.S. and China were selected as study areas for two key reasons. First, both regions provide access to extensive airborne LiDAR datasets: NASA's Land, Vegetation, and Ice Sensor (LVIS) (Blair et al., 1999) in the U.S., and small-footprint airborne LiDAR data in China. Second, the performance of forest height inversion can be assessed in a unique way: the New England region offers abundant GEDI calibration sites, while the northeastern part of China is situated at a comparable region at a similar latitude but without dedicated GEDI calibration sites.

Several experimental and validation sites are selected across the northeastern regions of the U.S. and China, with their relevant information summarized in Table 1 and Table 2, respectively. In the New England region, validation is conducted at various sites including the Howland Forest site in Maine, Harvard Forest site in Massachusetts, the White Mountain National Forest (WMNF) site in New Hampshire, the Green Mountain National Forest (GMNF) site in Vermont, and the Naugatuck State

Forest (NSF) site in Connecticut. These forest validation sites are covered by medium footprint (25 m) LVIS data acquired either in 2009 or in 2021. Particularly, the forest height inversion over the WMNF site was evaluated using GRANIT airborne laser scanning (ALS) data acquired in 2011 (Haans et al., 2009), with the canopy height product extracted from the waveform data at a raster sampling spacing of 2 m.

**Table 1 The forest validation sites covered by the airborne LiDAR observation in the New England, U.S**

| Validation sites | Location | Dominated tree species | LiDAR data acquisition year | Slope statistics | ALS validation area (ha) |
|---|---|---|---|---|---|
| Howland Forest | 68°44′ W, 45°12′N | Red spruce (Picea rubens Sarg.) and eastern hemlock | 2009 | Mean: 2.3° STD: 5.3 | $4.77 \times 10^4$ |
| Harvard Forest | 72°11′W, 42°31′N | Red oak, Red maple, Black birch, White pine, Eastern hemlocc | 2021 | Mean: 5.5° STD: 4.6° | $4.87 \times 10^4$ |
| White Mountain National Forest | 71°18′W, 44°6′N | Red Spruce, Eastern Hemlock, American Beech, and Red Maple, | 2011 | Mean: 9.7° STD: 8.6° | $1.20 \times 10^4$ |
| Green Mountain National Forest | 73°04′W, 43°57′N | Sugar maple, American beech, red maple, yellow and paper birch | 2021 | Mean: 10.4° STD: 7.6° | $8.91 \times 10^4$ |
| Naugatuck State Forest | 73°00′W 41°27′N | Northern red oak, Mixed upland hardwoods, Yellow-poplar | 2021 | Mean: 5.2° STD: 4.5° | $3.58 \times 10^4$ |

Regarding northeastern China, evaluation was performed at one forest site in each province: the Mengjiagang Forest site in Heilongjiang province, the Dagujia Forest site in Liaoning province, the Saihanba Forest site in Hebei province, the Hubao National Park in Jilin province, and the Genhe Forest Bureau in Inner Mongolia province. Validation of the forest height product across all the forest validation sites in China was done by comparisons with the small footprint (0.5-1 m) ALS data.

To provide a preliminary assessment of forest disturbances in these two regions, forest disturbance maps derived from global forest change products (Hansen et al., 2013) are shown in Figure 3 and Figure 4.

**Table 2 The forest validation sites covered by the ALS validation data in northeastern China**

| Validation sites | Location | Dominated tree species | LiDAR data acquisition year | Slope statistics | ALS validation area (ha) |
|---|---|---|---|---|---|
| Mengjiagang Forest site | 130°42′E, 46°25′N | Coniferous plantations (Larix gmelinii and Pinus syvestris) | 2017 | Mean: 6.6° STD: 5.6° | $3.78 \times 10^4$ |
| Dagujia Forest site | 125°00′E, 43°21′N | Coniferous plantations (Larix kaempferi, Pinus koraiensis, etc) | 2018 | Mean: 13.5° STD: 7.4° | $3.66 \times 10^4$ |
| Saihanba Forest site | 117°18′E, 42°24′N | Larix principis-rupprechtii, Pinus syvestris, and Betula | 2018 | Mean: 8.7° STD: 7.3° | $2.98 \times 10^4$ |
| Genhe Forest bureau | 121°32′E, 50°47′N | Larix gmelinii, Betula platyphylla, Populus davidiana | 2022 | Mean: 7.0° STD: 5.4° | $1.09 \times 10^5$ |
| Hubao Forest site | 130°12′E, 43°28′N | Mongolian oak, Basswood, Betula platyphylla | 2018 | Mean: 8.7° STD: 7.4° | $3.99 \times 10^5$ |

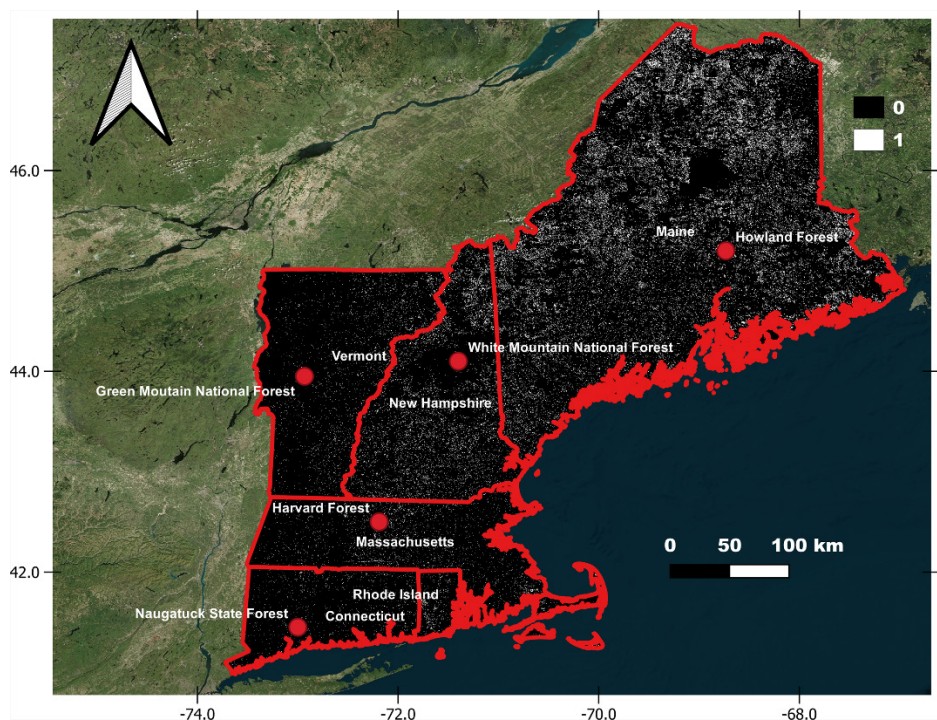

Figure 3: Forest disturbance map of the New England region (2007–2023) derived from the Global Forest Change dataset (Hansen et al., 2013). The binary classification distinguishes undisturbed areas (0) from disturbed areas (1) within the period.

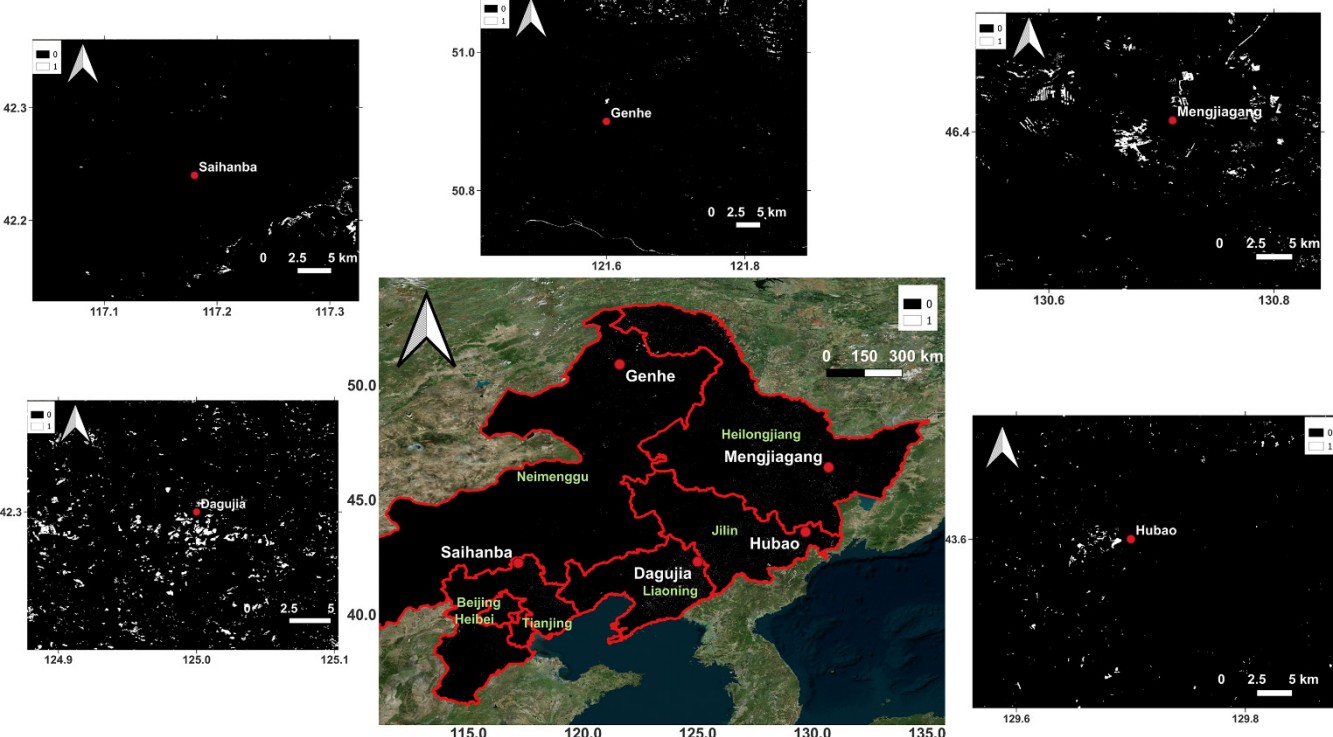

**Figure 4: Forest disturbance map of northeastern China (2007–2023) derived from the *Global Forest Change dataset* (Hansen et al., 2013). The binary classification distinguishes undisturbed areas (0) from disturbed areas (1) within the period. Province names are shown in green on the central map.**

## 2.2 Spaceborne and Airborne Remote Sensing datasets

The freely accessible L-band InSAR data from Japanese Aerospace Exploration Agency (JAXA)'s Advanced Land Observing Satellite (ALOS) mission were used for generating the InSAR correlation observations. In addition, a few radar data from ALOS-2 (a follow-up mission of ALOS-1) were also employed as case studies for validating the fusion of Radar-LiDAR under

205 concurrent conditions. Furthermore, the spaceborne LiDAR waveform-based metrics (rh98 as a proxy of forest height) from GEDI mission were used for parameterizing the temporal decorrelation model.

### 2.2.1 Spaceborne InSAR Datasets

Global Fine Beam Dual-polarization (FBD) SAR images with spatial grid of (10 × 3 m for range and azimuth directions) were collected by ALOS satellite from 2007 to 2010, with a repeat cycle of 46 days. To generate forest height mosaic, 100 cross-

210 polarized ALOS-1 InSAR scenes (identified by one pair of frame and orbit numbers) were processed to cover the New England region in the U.S., whereas more than 600 InSAR scenes were processed for covering the northeast of China. The InSAR

preprocessing was done by Jet Propulsion Laboratory (JPL)'s InSAR Scientific Computing Software (ISCE) software (Rosen et al., 2012). It was reported by (Lei and Siqueira, 2014) that the ALOS-1 InSAR observations acquired during the summer/fall time frame of 2007 and 2010 tended to have higher InSAR coherence. In the practical processing, multiple cross-polarized interferograms during the lifetime of ALOS-1 (2007-2011) were formed and processed for each scene based on different combinations of acquisition dates, allowing for the identification of the best InSAR pair for each ALOS-1 observation scene.

As a follow-on mission to ALOS-1, ALOS-2 had a shorter revisiting period (14 days), resulting in a better InSAR correlation behavior. The acquisition started from 2016, allowing for a nearly concurrent observations with respect to GEDI samples. However, the acquisition strategy and limited access to the high-resolution dual-polarized strip-map data have made it more difficult to form proper InSAR pairs and perform large-scale mapping. The grid of ALOS-2 image in FBD mode is at a grid of $8 \times 4$ m for range and azimuth directions. A windows size of $4 \times 8$ looks is used for the coherence estimation, resulting in an averaged pixel size of 30 m.

In this study, the use of limited ALOS-2 data is devoted to demonstrating the proposed approach under ideal case. The large-scale mapping capabilities were demonstrated using free-access ALOS data over the temperate forest regions. It is also noted that there is time discrepancy between the acquisition dates of ALOS and GEDI. This discrepancy is addressed using the twofold solution as detailed in Subsection 3.3. As for the abrupt discrepancy due to forest disturbance (e.g. logging, deforestation, and fire) that usually results in no/short vegetation with small backscatter values, replacing the InSAR inverted forest estimates with those derived from the appropriate ALOS-2 backscatter mosaic map for short vegetation (as shown in subsection 3.1) can detect the disturbed forest areas. And the study area is mainly concentrated on temperate and boreal forests, the heights of mature temperate forests (intact forest landscape) remain almost stable with slight change. Nevertheless, a simulation in Subsection 3.4.3 shows the approach of this study can approximate the height of the forests subject to natural growth at regional scale. In other words, all the InSAR based height inversions are calibrated to the acquisition time of GEDI, and thus best compared with the concurrent airborne LiDAR validation data.

### 2.2.2 Spaceborne LiDAR Datasets

As the first spaceborne LiDAR mission to characterize ecosystem structure and its dynamic, the NASA's GEDI mission was launched in December 2018. GEDI provides near-global measurements of forest structure metrics from 51.6° S to 51.6° N until 2024. With three lasers mounted, eight parallel tracks of samples at a footprint of 25 m are simultaneously collected. The spatial separation between samples during one datatake is 60 m in the along-track direction and 600 m in cross-track direction. The GEDI rh98 metric is selected as an appropriate proxy for indicating forest canopy height within each footprint because it has less sensitivity to errors as compared to the rh100 metric (Hofton et al., 2020). After filtering out GEDI samples with less

penetration sensitivities (e.g., 95% sensitivity, 50 m maximum elevation difference between GEDI and TanDEM-X

measurements), the remaining L2A version 2 GEDI samples are used for calibrating the inversion model.

### 2.2.3 Airborne LiDAR data

#### 2.2.3.1 Medium-footprint LiDAR data

A significant amount of airborne full-waveform LiDAR data was collected across the U.S. using LVIS sensor. The LiDAR

data were processed into rh98 maps at 25 m grid. Specifically, LiDAR data over the Howland Forest site in Maine were obtained in 2009. LVIS data acquired in 2021 for GEDI calibration cover all the other forest sites in the New England region, which were classified into four parts based on the state boundaries of New Hampshire, Vermont, Massachusetts, and Connecticut.

#### 2.2.3.2 Small footprint LiDAR data

In some sites, small footprint LiDAR data have to be used for validation. The validation at the WNMF site utilized GRANIT LiDAR data, which has a 2 m footprint and was acquired in 2011. All the forest heights in the northeastern China were validated using small footprint LiDAR data. The airborne LiDAR data, with an average point density of 6 pts/m² over Hubao National Park, were acquired using an airborne LiDAR system owned by the Chinese Academy of Forest Inventory and Planning. The

260 observations covering all other forest sites in the northeastern China were acquired during 2017-2022 using the airborne remote sensing system developed by the Chinese Academy of Forestry (Pang et al., 2016), which has an average point density of 12 pts/m². It should be noted that validating the forest height estimates against airborne LVIS observations is not straightforward, as the footprint of these airborne data is much smaller than the footprint of GEDI. To address this footprint difference, an equivalent RH98 metric (referred to as ERH98 hereafter) needs to be extracted at the position of the 98th percentile of the

265 LiDAR waveform or from the histogram formed by high-resolution CHM estimates within a 25 m footprint. Following this procedure, all small-footprint LiDAR data were reprocessed to generate forest height estimates based on the ERH98 metric with the same footprint size.

### 2.2.4 Forest and Non-Forest Maps

Non-forest areas including water bodies and urban areas were masked out using the 2021 National Land Cover Database (Homer et al., 2015) and the 2021 ESRI Global Land Cover Map (Karra et al., 2021).

**2.2.5 Backscatter mosaic map**

This study used the global radar backscatter products generated by JAXA using ALOS-1/2 FBD images (Shimada et al., 2016) after radiometric and geometric calibration (including slope effects correction). Specifically, the global cross-polarized backscatter products from 2019 and 2020 over the northeast of the U.S. and China were utilized to obtain height estimates of short vegetation. These two-year products were used to account for missing data gaps, backscatter calibration inconsistencies, and to best match the acquisition time of the validation airborne LiDAR data.

**3 Methodology**

This section first outlines the processing workflow and implementation steps, then provides a detailed analysis of the methodological principles.

3.1 Processing workflow

The entire processing workflow is illustrated in Figure 5. Figure 6 provides a zoomed-in example of global-to-local forest height inversion over the Howland Forest site. The standard interferometric preprocessing (including coregistration, topographic phase compensation, interferometric formation, and geocoding steps) for multiple ALOS-1 InSAR pairs is performed by using JPL's ISCE software, in which the parallel computing capabilities of Graphic Processing Units (GPUs) is utilized to enhance the processing efficiency, generating the InSAR coherence map as shown in Figure 6 (a). After matching the GEDI samples onto the grid of InSAR scene (the collocated GEDI rh98 samples are shown in Figure 6 (b)), the global-to-local inversion for each scene is carried out as follows:

1. **Global Fitting:** As illustrated in subsection 3.3, a global fitting is first performed for obtaining an initial guess of temporal parameters $(S_0, C_0)$ (see Equation (3)) using all the available GEDI rh98 samples and corresponding InSAR observations over each InSAR scene. As noted earlier that a moderate forest disturbance may result in an overestimation of forest height for global fitting (Lei et al., 2017b), a re-weighted iterative global fitting is instead used to remove the gross errors induced by forest disturbances after the first iteration.

2. **Local fitting:** A local fitting is then conducted around each GEDI sample within a local boxcar window with a constraint of smaller searching range in the vicinity of the initial estimates, i.e., $(S \in [S_0 - r_S, S_0 + r_S], C \in [C_0 - r_C, C_0 + r_C])$ where $r_S$ and $r_C$ represent the searching region for two parameters, respectively. Spatial distance-based weights (e.g., Figure 7) are preferred in local fitting for preserving detailed information and ensuring the robust inversion over large-scale application. An expensive computational burden is implied for the local fitting so that a GPU-based implementation (Yu et al., 2019) is developed for enhancing processing efficiency.

3. **Interpolation and inversion**: the irregularly distributed temporal decorrelation parameters i.e., $\{S(i,j), C(i,j)\}$ with $i, j$ denoting the latitude and longitude for each GEDI sample, are interpolated into the regular grid of InSAR

coherence magnitude map based on the Delaunay-triangulation based natural neighborhood interpolation. The forest height is then inverted on a pixel-by-pixel basis using the InSAR coherence observation and the inversion model

(generating wall-to-wall forest height mapping as shown in Figure 6 (c)).

4. **Backscatter-based estimates**: The forest height estimates of short vegetation in the previous inversion are replaced by the corresponding backscatter-based estimates (see Figure 6 (d)): the GEDI rh98 samples over bare ground and short vegetation landcovers (i.e., $0 \leq rh98 \leq 10$ m) along with the corresponding backscatter information are fitted using an exponential model, which is then used to obtain backscatter-to-height estimates (Lei et al., 2019). Short

trees are identified using a criterion where backscatter-derived forest height estimates fall below 10 meters, based on the maximum height of shrubs (Lawrence, 2013; Allaby, 2012) and empirical studies (Lei et al., 2019).

5. **Mosaicking**: a mosaicking approach is finally carried out to pick up the best InSAR pairs based on the pre-inversion metric and mosaicking the overlapping area using the pixels with better goodness of fit as discussed in Subsection 3.5.

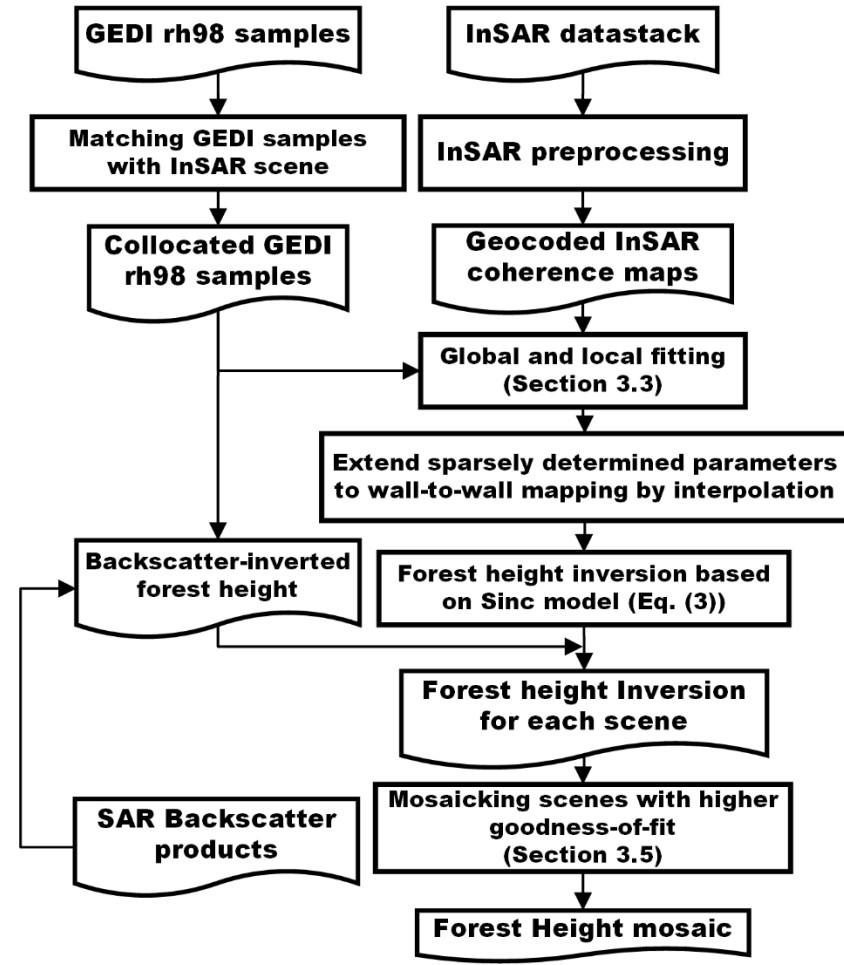

**Figure 5: Block diagram of the workflow for generating forest height mosaics.**

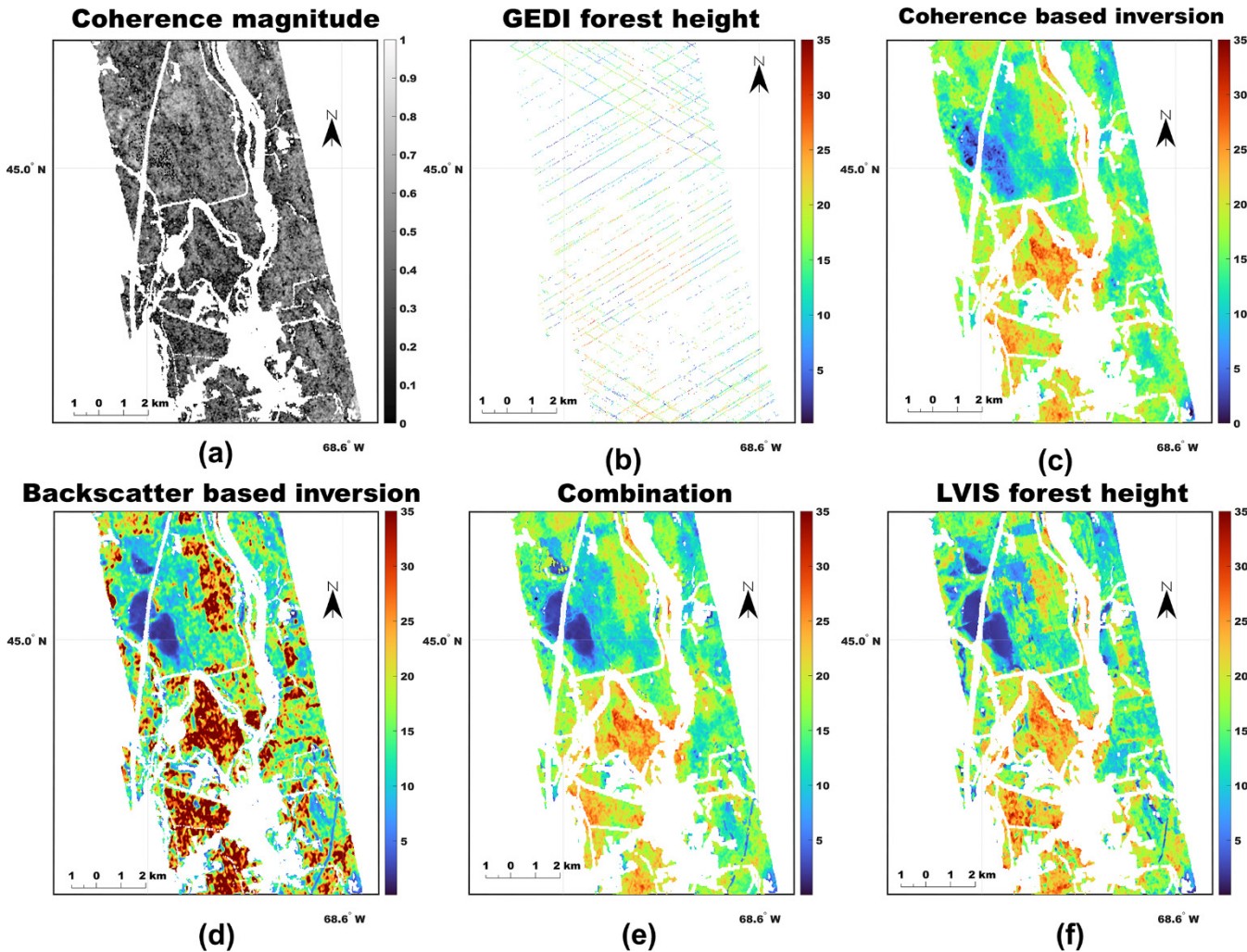

**Figure 6 An illustrative example of the processing steps at the Howland Forest site: (a) the input ALOS-1 coherence magnitude map; (b) the GEDI rh98 samples; (c) the forest height estimates based on InSAR coherence information; (d) the backscatter based height estimates; (e) the final forest height map after replacing the estimates of short trees in (c) with the collocated pixels in (d); (f) is the airborne LVIS LiDAR validation data.**

### 3.2 *Sinc* Inversion model

After standard InSAR preprocessing (including coregistration, and topographic phase compensation, etc), a complex InSAR correlation observation between two SAR images can be derived by:

$$\gamma = \frac{\langle I_1 \cdot I_2^* \rangle_L}{\sqrt{\langle I_1 \cdot I_1^* \rangle_L \langle I_2 \cdot I_2^* \rangle_L}} \tag{1}$$

where $I_1$ and $I_2$ represent two SAR images from the 1st and 2nd acquisition, respectively, the operator $\langle \cdot \rangle_L$ is used for spatial averaging of $L$ looks. As a metric to measure similarity, the complex InSAR correlation accounts for various forms of decorrelation (Zebker and Villasenor, 1992; Gatelli et al., 1994; Treuhaft and Siqueira, 2000). If we consider a pair of SAR images with short spatial separation over forested area, InSAR coherence can be expressed as:

$$|\gamma| = |\gamma_{SNR}| \cdot |\gamma_{geo}| \cdot |\gamma_{v\&t}| \tag{2}$$

where $\gamma_{SNR}$ represents the decorrelation induced by the radar-return signal-to-noise ratio, and $\gamma_{geo}$ denotes the geometric decorrelation due to the difference between two look angles. After accounting for SNR decorrelation and performing common band filtering (Gatelli et al., 1994), the remaining component of correlation, $\gamma_{v\&t}$, is only related to temporal changes and the distribution of scatterers in the vertical direction. The spatial separation for one InSAR pair is usually indicated by the interferometric vertical wavenumber $\kappa_z$ (unit rad/m) (Bamler and Hartl, 1998). A small value of $\kappa_z$ (less than 0.05 rad/m) is
usually suggested by the analysis in (Lei and Siqueira, 2014).

For moderate and large temporal baselines, moisture-induced dielectric fluctuations and wind-induced random motion are identified as the primary factors influencing temporal decorrelation (Lavalle et al., 2012; Askne et al., 1997). (Lei et al., 2017b) introduced a modified Random Volume over Ground (RVoG) model that accounts for the coupled effects of dielectric changes
and random motion in spaceborne ALOS-1/2 observations. The model is further simplified by assuming negligible ground scattering under cross-polarized (HV) observations and short spatial baselines, and establishes a semi-empirical formula linking cross-polarized InSAR coherence magnitude ($\gamma_{v\&t}^{HV}$) to forest height ($h_v$):

$$|\gamma_{v\&t}^{HV}| = S \cdot \mathrm{sinc}\left(\frac{h_v}{C}\right), \text{with } 0 \leq h_v \leq \pi C \tag{3}$$

where $S$ (ranging from 0 to 1; unitless) is a parameter primarily connected to moisture-induced dielectric changes in the target, and $C$ (a positive value that has units of meters) relates to the wind-induced random motion. In practice, ground scattering
remains present in cross-polarized signals, particularly for low-frequency SAR observations. However, it is typically minimal, with a ground-to-volume ratio generally less than 0.1. This residual ground scattering can bias forest height estimates, particularly for short or very tall forest stands. It is also important to note that this formula is valid when the interferometric vertical wavenumber is below 0.15 rad/m and is most reliable when smaller than 0.05 rad/m (Lei and Siqueira, 2014), which is consistent with the acquisition geometry of ALOS-1/2 InSAR acquisition. Otherwise, the presence of volumetric
decorrelation will result in a compromised inversion performance.

### 3.3 Fusing concurrent radar-lidar acquisitions by a global-to-local inversion approach

Determination of the above temporal decorrelation model (3) has to resort to ancillary forest height data e.g., field inventory data, LiDAR measurements, and etc. A global-to-local procedure is used for determination, since the regional forests are

usually relatively homogeneous, leading to the construction of ill-conditioned observations for model determination. A global initial guess is firstly needed for constraining the solution space for subsequent fitting.

For one InSAR scene, a constant scene-wide behavior of the temporal change parameters is firstly assumed, e.g., $(S_{scene}, C_{scene})$. To one candidate pair of these parameters, forest height estimates $\widehat{h_v}$ are derived by solving the model (3) based on InSAR coherence magnitude. A covariance matrix between $\widehat{h_v}$ and the ancillary forest height $h_a$ from LiDAR and its eigen-decomposition are expressed by:

$$\Sigma = \begin{bmatrix} \text{cov}(\widehat{h_v}, \widehat{h_v}) & \text{cov}(\widehat{h_v}, h_a) \\ \text{cov}(\widehat{h_v}, h_a) & \text{cov}(h_a, h_a) \end{bmatrix} = Q\Lambda Q^{-1} \tag{4}$$

where the 2×2 matrix $Q$ contains the eigen vectors, the 2×2 diagonal matrix $\Lambda$ comprises the eigen values. The function $\text{cov}(A, B)$ denotes the covariance between the two vectors $A$ and $B$:

$$\text{cov}(A, B) = \frac{1}{N-1} \sum_{i=1}^{N} (A - \mu_A) \cdot (B - \mu_B) \tag{5}$$

with $\mu_A$ and $\mu_B$ being the mean values of two input vectors. Two fitting metrics e.g., slope $k$ and bias $b$ can be extracted out of the constructed covariance matrix:

$$k = \frac{Q_{21}}{Q_{11}} \tag{6}$$

$$b = 2\frac{\mu_{\widehat{h_v}} - \mu_{h_a}}{\mu_{\widehat{h_v}} + \mu_{h_a}} \tag{7}$$

By defining a figure of merit i.e. approaching zero bias and unity slope (i.e. the 1:1 line), a pair of temporal parameters are determined by minimizing the objective function:

$$\{S_{scene}, C_{scene}\} = \arg \min_{S,C}(b^2 + (k - 1)^2) \tag{8}$$

Resolving model parameters $(S_{scene}, C_{scene})$ is referred to as global fitting strategy. The effectiveness of this approach was demonstrated over the northeastern U.S. (Lei et al., 2018) if well-constructed observation vectors including sufficient samples ranging from low trees to tall trees[1].

Because temporal change factors tend to be spacing-varying with respect to vegetation types and weather/climate conditions, inversion performance for solely global fitting as described above is expected to deteriorate when applied for large-scale inversion. With the availability of GEDI data, a substantial improvement is expected by expressing decorrelation model at a finer spatial scale. A preliminary effort was made in (Lei and Siqueira, 2022) where the determination of $(S, C)$ is carried out on each GEDI LiDAR sample $h_{gedi}(i, j)$ to obtain $S(i, j)$ and $C(i, j)$, where $i$ and $j$ represent latitude and longitude for each

---

[1] This fitting metric has proven to be more robust than the Euclidean norm for a data cloud with large measurement uncertainty (Lei and Siqueira, 2014).

GEDI sample. As only one sample cannot provide sufficient observations to resolve two unknowns so that an external constraint on the ratio of $S(i,j)/C(i,j)$ was introduced.

Each individual GEDI measurement is subject to errors caused by artifacts such as terrain slopes (Wang et al., 2019) and systematic geo-location inaccuracies ranging from several meters to tens of meters (Tang et al., 2023). Additionally, the penetration of the LiDAR signal is sometimes limited within GEDI's coverage beams. InSAR coherence estimates also experience measurement uncertainty (Rodriguez and Martin, 1992; Touzi et al., 1999). To address these challenges, spatial averaging is commonly used to reduce errors for both coherence and GEDI measurements.


In this context, a local fitting step is introduced to acquire temporal change factors at a finer spatial scale using extensive GEDI information: a circular window (32 pixels, 960 m wide) is set for each GEDI sample to collect regional samples and their corresponding InSAR coherence magnitude observations. A Euclidean norm-based fitting is used since a local window tends to encompass homogeneous vegetation with similar heights so that the global $k$-$b$ fitting metric is no longer robust. Assuming

consistent temporal change factors within each local window, the factors can be determined by:

$$\{S(i,j), C(i,j)\} = \arg\min_{S,C} \sum_{(r,c)\epsilon W} \left(\widehat{h_v}(i+r, j+c) - h_{gedi}(i+r, j+c)\right)^2 \tag{9}$$

where $W$ is the local searching window in geographic coordinate system with $r$ and $c$ being local indices along the latitude and longitude direction within the local window. To preserve detailed information, weighting factors based on distance or similarity metric are used in local fitting, as in:

$$\{S(i,j), C(i,j)\} = \arg\min_{S,C} \frac{\sum_{(r,c)\epsilon W}\left[w(r,c)\left(\widehat{h_v}(i+r,j+c) - h_{gedi}(i+r,j+c)\right)\right]^2}{\sum_{(r,c)\epsilon W}(w(r,c))^2} \tag{10}$$

where the weights can be set based on spatial distances between neighboring pixels and the central pixel within the local window, or adapted depending on land-cover type (Deledalle et al., 2014). For computational efficiency and versatility, the spatial distance-based weight setting, as exemplified in Figure 7, has been used for inversion in this work. The optimal local window size is determined by minimizing the root mean square error (RMSE) of forest height estimates for moderate-to-tall

trees (>10 m) across diverse window size configurations, validated against independent LiDAR datasets. The selection of window size is always a compromise between smooth and detailed information. This window size is selected here for including enough samples for model fitting while maintaining local detailed information. As the parameters were determined on the grid of GEDI samples, a post-interpolation is needed to obtain gridded temporal parameters for matching the InSAR scene.

It should be noted that this physical scattering model (3) is established for forested areas, meaning it does not adequately address other land-cover types, such as water bodies and urban areas. Furthermore, actively managed regions (e.g., cropland) are subject to additional decorrelation effects induced by human activities. It was suggested to use L-band backscatter-based height estimates to improve the inversion in these instances. The backscatter-based inversion are derived based on an

exponential model (Yu and Saatchi, 2016; Lucas et al., 2006) which can be determined using the similar global-to-local routine

as the proposed approach. In practice, only global fitting strategy is used for model parameterization as it provides accurate estimates as compared to global-to-local fitting (slightly worse) while maintains an affordable computational expense. After that, the estimates of active land management areas in coherence-based inversion are replaced by the corresponding backscatter-based estimates.

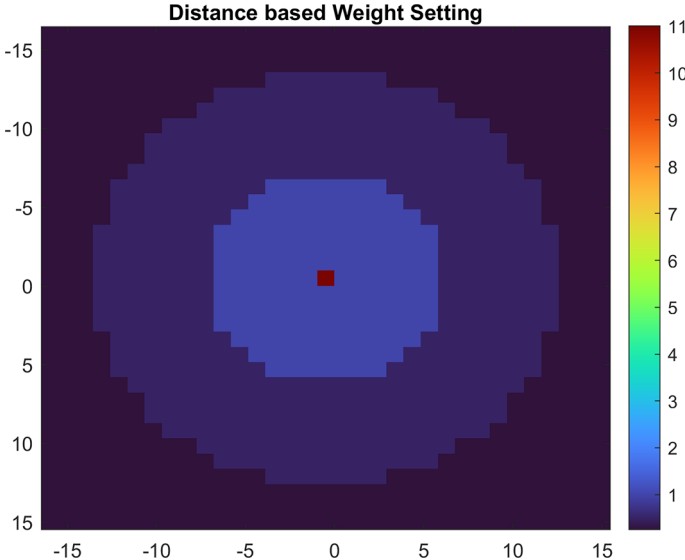

**Figure 7: An illustrated example of distance-based weight setting for a 0.96 km (32 by 32 pixels) wide moving boxcar window.**

### 3.4 Fusing non-concurrent Radar-LiDAR data

### 3.4.1 Addressing Forest disturbance or Deforestation

The acquisition times between repeat-pass InSAR and spaceborne LiDAR usually do not overlap in practice. For example, the

ALOS-2 InSAR acquisitions would match those of GEDI ideally; however, there are not many ALOS-2 InSAR pairs available in the data archive that are freely accessible. In contrast, ALOS InSAR pairs are consistently available covering the study regions; however, there is an approximate 10-year time gap. To address this issue, a twofold solution is provided in Section 3. The temporal evolution of forests is primarily influenced by two factors: forest disturbance (including deforestation) and natural growth. Forest disturbance leads to forest loss and, in some cases, conversion to other land uses, such as bare ground.

This change is poorly characterized by the signal model (3) and must be addressed in advance.

The height of bare ground and short vegetation ($\leq 10\ m$) (Lawrence, 2013; Allaby, 2012) is better inverted by jointly using the SAR backscatter and spaceborne LiDAR data compared to the InSAR correlation-based approach (Lei, 2016). Moreover, annual global backscatter products are available from the archive of JAXA, enabling the fusion of the ALOS-2 SAR backscatter

information and spaceborne LiDAR under nearly concurrent acquisition conditions. Figure 8 presents an example of how demonstrates forest loss detection by SAR backscatter-based inversion within the New England region: the majority of forest disturbance areas as defined by the global forest change products (Hansen et al., 2013) were detected by the backscatter-based estimates. Based on the statistics over the New England region, 72% of disturbed areas have been detected using the backscatter-based estimates.


Short forests undergoing rapid temporal height variations within short intervals cannot be adequately captured by current ALOS-1/2 datasets. These dynamic changes can be better resolved using dense time-series data from TanDEM-X (Treuhaft et al., 2017; Lei et al., 2018), Sentinel-1 (Bhogapurapu et al., 2024), and the forthcoming NISAR mission.

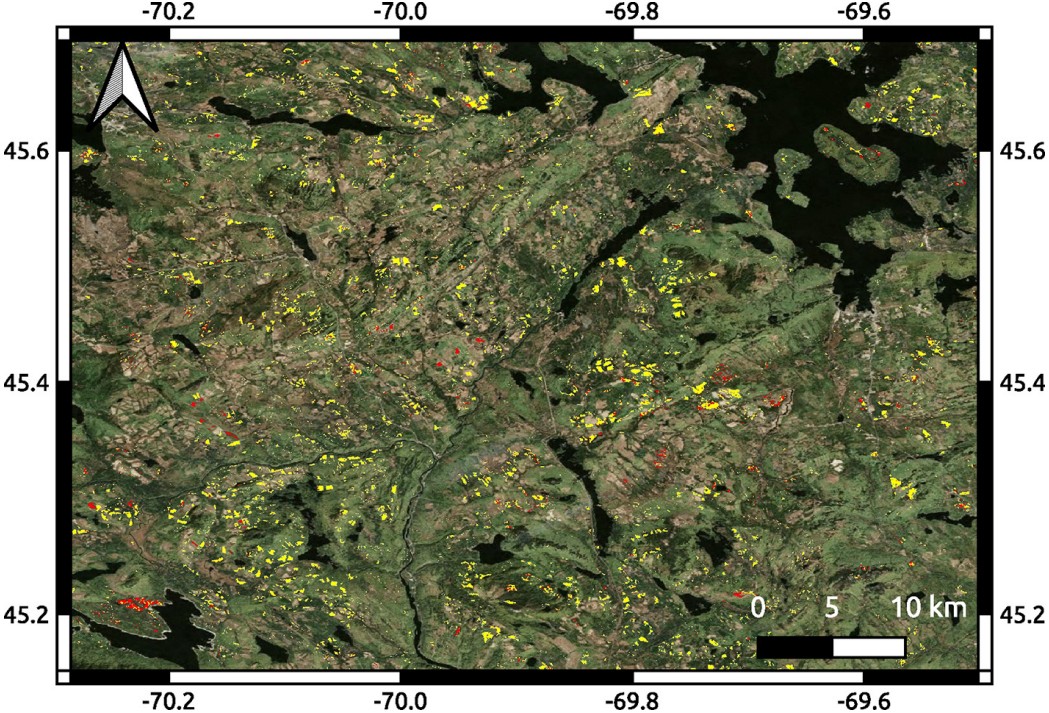


**Figure 8: Example of fusing the ALOS-2 backscatter products and GEDI to detect forest disturbances as defined by Forest Change products (Hansen et al., 2013) over a representative region in the new England region. Yellow pixels represent forest disturbance areas detected by backscatter-based estimates, while red pixels indicate disturbed areas as not identified by these estimates.**

### 3.4.2 A modified model considering the natural forest growth

The natural growth of forests remains another concern. In temperate forest regions, the Intact Forest Landscape (IFL) along with forest height maintain quite stable and only change slightly as trees grow (Potapov et al., 2008; Riofrío et al., 2023). In the New England region, the growth rate was found to be inversely proportional to forest height. This finding is based on a

comparison between ICESat-1 or LVIS LiDAR data acquired before 2009 with LVIS LiDAR data collected in 2021. Using these datasets, the temporal evolution of forest height at two epochs is modelled as follows:

$$h_v(t_2) = h_v(t_1) + G \cdot (t_2 - t_1) \tag{11}$$

where $h_v$ represents the time-dependent forest height, $t_1, t_2$ denote the initial and subsequent epochs, respectively. Although the forest growth rate is derived by comparing pre- and post-growth states, it can be modelled based only on forest height data from either epoch, for example, using the initial forest height $h_v(t_1)$:

$$G = a \cdot h_v(t_1) + b \tag{12}$$

Where $a$ and $b$ are linear coefficients. If a dense time-series of forest height data over certain forest land cover is provided, the above equation can be constructed in a differential form as:

$$\frac{\partial h_v}{\partial t} = a \cdot h_v + b \tag{13}$$

This ordinary differential equation yields the following expression:

$$H(t_1) = \left( \frac{b}{a} + H(t_2) \right) e^{-a(t_2 - t_1)} - \frac{b}{a} \tag{14}$$

This expression aligns with the Hossfeld model. However, it requires statistical data on annual growth rates, which is often unavailable on a large scale in practice. When the parameter $a$ is small, Eq. (14) simplifies to the form presented in Eq. (12). Evaluation in the New England region indicates the absolute value of $a$ is less than 0.02. By substituting Equation (12) into the signal model, the following modified model is obtained

$$|\gamma_{t\&v}^{HV}(t_1)| = S(t_1) \cdot \text{sinc} \left( \frac{h_v(t_2) - b \cdot dt}{(1 - a \cdot dt) \cdot C(t_1)} \right) \tag{15}$$

where $|\gamma_{t\&v}^{HV}(t_1)|$ represents the InSAR coherence at initial time $t_1$, it follows the model is shifted and scaled with respect to the original model (3), $dt = t_2 - t_1$.

As a representative example, Figure 9 illustrates the fitted forest growth rate ($G = -0.0134 \cdot h_v(t_1) + 0.464$; unit: $m/year$) for the forest height interval with the highest density at the Harvard Forest site in Massachusetts. This analysis compares LVIS 465 LiDAR data acquired in 2009 and 2021, after filtering out disturbed forests areas ($G \leq -0.1 \, m/year$) and short vegetation ($h_v \leq 10 \, m$). To illustrate how the InSAR correlation observations from ALOS data are linked to GEDI forest estimates (with time gap of around 10 years), a simulation can be performed by inserting the fitted growth rate into the model (15) and setting $S(t_1) = 0.9$ and $C(t_1) = 11$. Figure 10 presents the simulated results of original sinc model and its modified version. It follows that the forest height below $b \cdot dt$ cannot be well characterized. This value usually ranges from 8 to 15 in the investigation 470 over the New England region. This finding also highlights the importance of utilizing backscatter information to estimate the heights of short forests in this case. We remark that natural growth functions are highly species-specific. From a practical inversion perspective, the application of the modified model requires precise detailed statistics of natural growth across various forest types on a large scale. However, such data are currently unavailable, as existing spaceborne LiDAR datasets lack

collocated measurements from two distinct time periods. In the absence of comprehensive forest growth data, this model is not yet recommended for direct large-scale use. Instead, it can be integrated into the framework of the original model by adjusting temporal parameters. The following subsection provides simulation examples to demonstrate this adaptation.

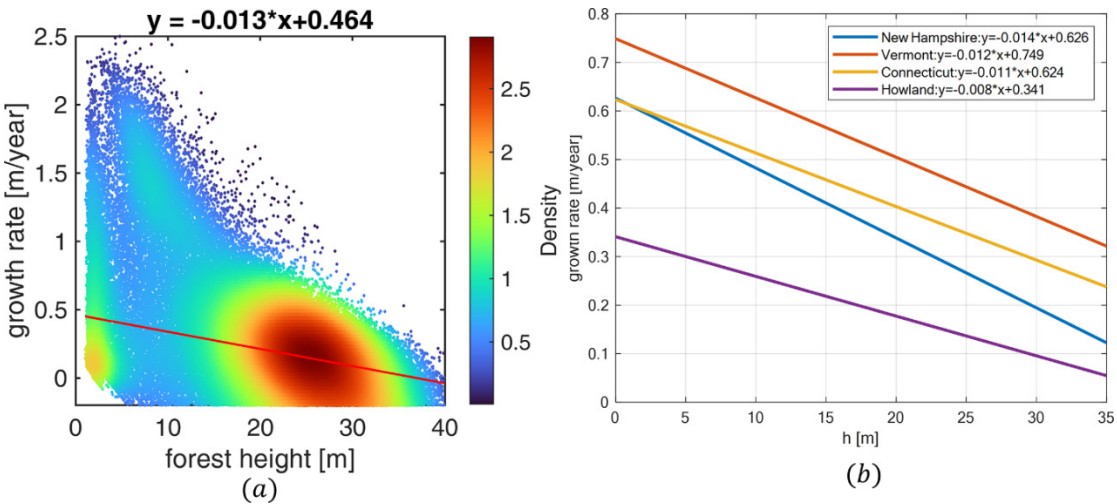

(a)  (b)

Figure 9: (a) Scatterplot of forest height growth rate versus forest height, derived from a comparison of LVIS forest height estimates between 2009 and 2021 at the Harvard Forest site, Massachusetts. The red line represents the fitted growth rate function of forest height in 2009 over the high-density region. (b) the fitted forest height growth rate functions at typical forest sites of other states in the New England region.

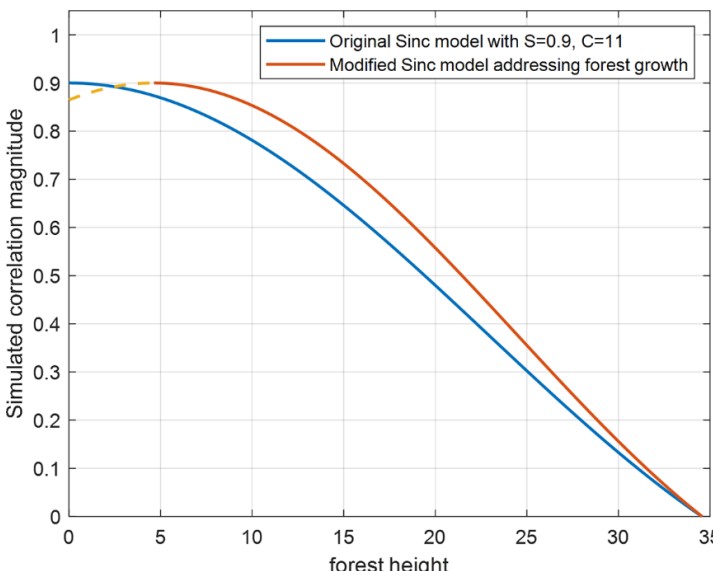

Figure 10: Simulation results comparing the original sinc model and its modified version incorporating natural forest growth at the Harvard Forest site (with forest growth rate function of $G = -0.0134 \cdot h_v(t_1) + 0.464$). The blue line represents the original sinc model with established temporal parameters, while the red line shows the modified model accounting for forest height growth. The dashed orange line highlights the region that cannot be characterized by the modified model. The y-axis represents the coherence magnitude estimated from Equation (3) and (15).

### 3.4.3 Approximating the modified model with the original model in the absence of natural growth data

Without detailed forest growth statistics, this subsection demonstrates the modified sinc model can be well approximated in the framework of the original sinc model but with updated parameters $(S', C')$ using simulation. This enables large-scale application achieved in the framework of original model without detailed growth statistics.

It is almost impossible to derive $S'$ and $C'$ based on a direct one-to-one analytical transformation between the models (15)

and (3). Instead, such parameters can be determined by aligning the original model and modified model at two fixed points. For example, Figure 11 presents the resolved parameters when the alignment is performed for two points at 10 and 30 m for global fitting at three typical forest sites (e.g., scenes with large forest height ranges). The best approximation is observed at the Howland Forest site where the forest height changes slowly. Model mismatches are observed at 20 m at other two Forest sites. This can be addressed by aligning two models for each short interval. Figure 12 presents the behavior of newly fitted

parameters when aligning either a short forest interval (e.g., 10–15 m) or a tall forest interval (e.g., 25-30 m). The results suggest that the modified sinc model is better approximated through piecewise fitting, emphasizing the importance of local fitting to approximate modified models for a short height range within relatively homogenous forest areas. Notably, $S'$ increases and even exceeds 1, particularly for taller trees, deviating from its original physical definition $S' \leq 1$ (e.g. only due to moisture-induced dielectric decorrelation in the original model). As seen in the behavior of the fitted parameters for each

short interval $[x\ x+5]$ in Figure 13, the $C'$ parameter is larger than the original parameter $C$ for short vegetation but approach $C$ as vegetation height increase. While the $S'$ parameter is close to the original $S$ parameter for short trees and becomes biased for tall trees.  For the three forest sites analysed, the White Mountain National Forest shows a larger forest height change rate, leading to a greater deviation between the fitted parameters $(S', C')$ and the original parameters $(S, C)$.

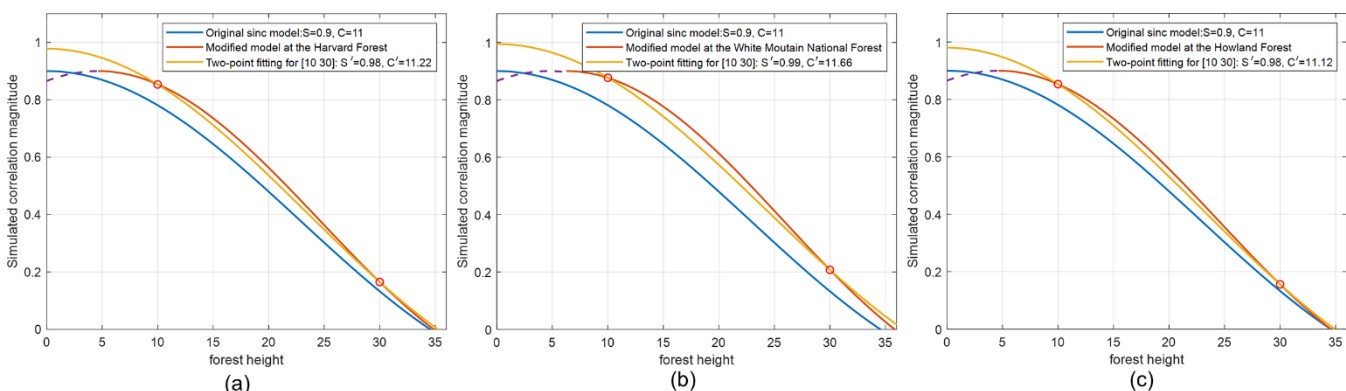


**Figure 11: Approximating the modified sinc model with the original model by aligning two points at 10 and 30 m: the newly fitted parameters at (a) Harvard Forest sites; (b) the forest site in Vermont; (c) Howland Forest site. The y-axis represents the coherence magnitude estimated from Equation (3) and (15).**

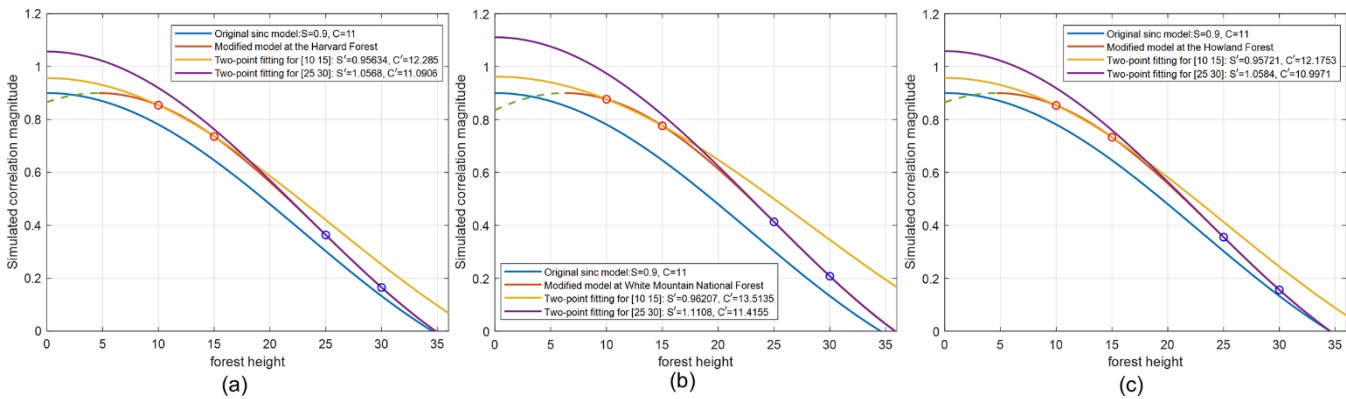

Figure 12: Approximating the modified sinc model with the original model by aligning two points either at 10 and 15 m or at 25 and 30 m: the newly fitted parameters at (a) the Harvard Forest site; (b) the White Mountain National Forest site; (c) the Howland Forest site. The y-axis represents the coherence magnitude estimated from Equation (3) and (15).

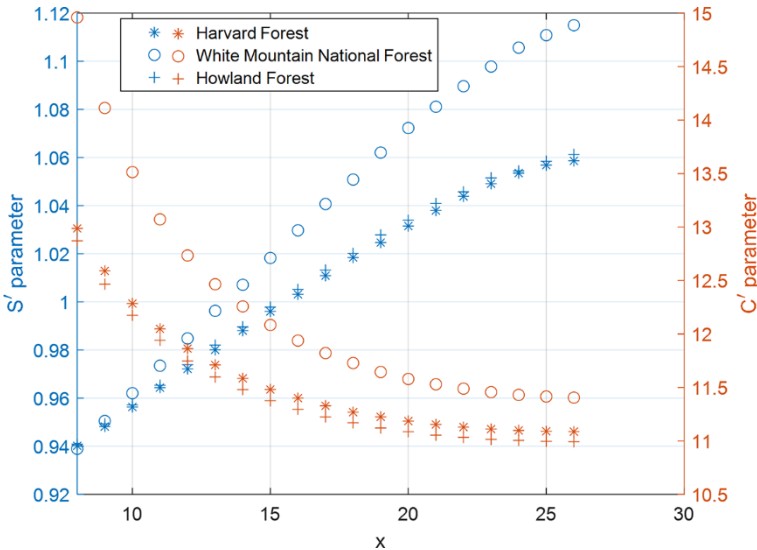

Figure 13: illustration of the dynamic behavior of fitted $(S, C)$ parameters for three representative forest sites when approximating the modified sinc model with original model by local-fitting at fixing points at each forest height interval $[x, x+5]$.

3.5 The selection of proper InSAR pairs and mosaicking

Due to several decorrelation factors (induced by precipitation, human activities, etc.), the temporal decorrelation behavior over forested scenarios is complicated in the context of repeat-pass InSAR. For example, total decorrelation is possible to occur for ALOS-1 InSAR pairs with a temporal span larger than two months during the monsoon season. In this case, the InSAR correlation behavior is dominated by additional decorrelation sources and hence would no longer be well-suited for forest height inversion.

In this context, a pre-inversion metric and a post-inversion error assessment need to be defined when multiple InSAR pairs are available, to eliminate those scenes that would not work well for forest height inversion.


An example of this evaluation is illustrated in Figure 14 and Table 3 below. Here, the fitness of the temporal change model (3) for a given InSAR scene can be evaluated by testing the underlying assumption of the physical scattering model. Specifically, taller trees are more easily decorrelated over time due to the larger deviation of random motions compared to the short ones. A simple yet effective pre-evaluation metric can be attained by linear regression between the GEDI rh98 samples

(only keeping forested land cover) and the coherence magnitude observations. Negative slope in the linear regression usually indicates a relevant validity of the inversion model. As shown in Figure 14 and Table 3, InSAR pairs with a negative slope tend to yield more accurate estimates (e.g., the image pair with slope of -4.76 gives better estimates with respect to the image pairs with slopes higher than -3). Note that while InSAR pairs with short temporal baselines may present higher correlation values, they may occasionally not be well-suited for the temporal decorrelation model due to regional precipitation.


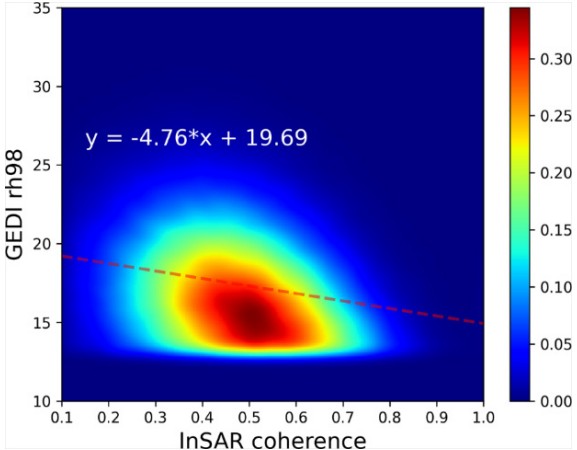

**Figure 14: Density scatterplot illustration of relation between repeat-pass ALOS-1 InSAR coherence (3×10 window at 30 m grid) and GEDI rh98, and a red dashed fitted line with the slope as the pre-inversion metric.**

**Table 3: An example to illustrate the pre-inversion metric (slope) and their inversion performance (Root Mean Square Error (RMSE) and coefficient of determination (R2)) for the available InSAR pairs in 2007 at the Howland Forest site in Maine, U.S.**

| InSAR pairs (time vs time) | Slope | RMSE | R2 |
|---|---|---|---|
| 20070710-20070825 | -2.82 | 4.09 | 0.45 |
| 20070825-20071010 | -2.57 | 4.14 | 0.45 |
| 20070710-20071010 | -4.76 | 3.8 | 0.47 |

The post-inversion metric is defined by using the figure of merit during the local fitting (see Equation (10)): once one pair of temporal parameters $(\{S(i,j), C(i,j)\})$ is determined, a weighted squared summation of the differences between inverted height estimates and GEDI measurements over a regional window is given by:

$$\varepsilon(i,j) = \frac{\sum_{(r,c)\epsilon W}\left[w(r,c)\left(\hat{h}_v(i+r,j+c)-h_{gedi}(i+r,j+c)\right)\right]^2}{\sum_{(r,c)\epsilon W}w(r,c)^2} \tag{16}$$

It can be noted that the derived post-inversion metric $\varepsilon(i,j)$ is also in the same coordinate system as GEDI samples. A Delaunay-triangulation based natural neighborhood interpolation (Park et al., 2006) can be used afterwards to attain pixel-by-pixel evaluation. This assessment serves to guide the mosaicking of overlapping regions between consecutive InSAR scenes by preserving estimates with lower $\varepsilon(i,j)$.

It should also be noted that since there are not enough InSAR pairs each year for ALOS-1 data, in this work, we only use the above-mentioned pre- and post-inversion metrics to select the best InSAR pair. However, if there are sufficient pairs from more recent spaceborne repeat-pass InSAR missions (such as 12 days for NISAR, 14 days for ALOS-4, and 4-8 days for LuTan-1), a synthetic InSAR coherence map can be generated by applying the monthly, seasonal median, or maximum operations (Kellndorfer et al., 2022).

## 4 Results

This section begins by presenting large-scale forest height mosaic maps for the northeastern regions of the U.S. and China, and is followed by extensive validation for representative individual forest sites.

## 4.1 Forest Height mosaic generation

The proposed inversion approach was developed as an automated open-source software, serving as version 2 of the Forest Stand Height (FSH) software (https://github.com/Yanghai717/FSHv2).

For generating the forest height mosaic map over the New England region (a total area of 18 million hectares), over 100 ALOS-1 InSAR scenes were processed using a multi-look averaging with two range looks and ten azimuth looks, leading to a pixel size of 20 by 30 m consistent with SRTM grid. Approximately 15 million GEDI rh98 samples were used for model parameterization. The height estimates of short vegetation were replaced with the backscatter-based estimates using the ALOS-2 backscatter products in either 2019 or 2021. A few ALOS-2 InSAR pairs were also used for demonstrating Radar-LiDAR

fusion under concurrent acquisitions. Non-forest areas were masked out based on the 2021 NLCD products. The mosaic was projected to the same geographic coordinate grid of SRTM DEM product. The forest height mosaic is depicted in Figure 15. The absence of discontinuity between adjacent scenes confirms the consistency of the forest height estimates. Additionally, the coastal region is included in the inversion, despite potential challenges posed by weather conditions (as reported in (Lei et al., 2019)). This underscores the advantage of using the global-to-local two-stage inversion approach to handle fast spatially-

varying temporal change factors induced by different land covers or weather/climate conditions. Further quantitative evaluation is conducted in subsequent sections, with a focus on each individual forest site.

For the northeast of China, 688 ALOS-1 InSAR scenes and 160 million GEDI samples were used to generate the mosaic covering the five provinces (total area of 152 million hectares). Non-forested areas were masked out based on the 2021 ESRI

global land cover maps. ALOS-2 global backscatter maps for 2019 and 2020 were employed to estimate the height of short trees to match the acquisition time of the validation airborne LiDAR data. The final forest height mosaic is shown in Figure 16. It is noted the area outside the coverage of GEDI observation (>51.6°N) were discarded.

The generated products are made available via https://doi.org/10.5281/zenodo.11640299 (Yu and Lei, 2024). Further

evaluation is shown in Subsections 4.2 and 4.3. In both cases, small values of $\kappa_z$ are maintained for all available InSAR pairs ($\kappa_z$ are below 0.15 rad/m, and the mean values are 0.032 rad/m and 0.029 rad/m for Chinese and American datasets, respectively) which is conform to the assumption made in (Lei and Siqueira, 2014).

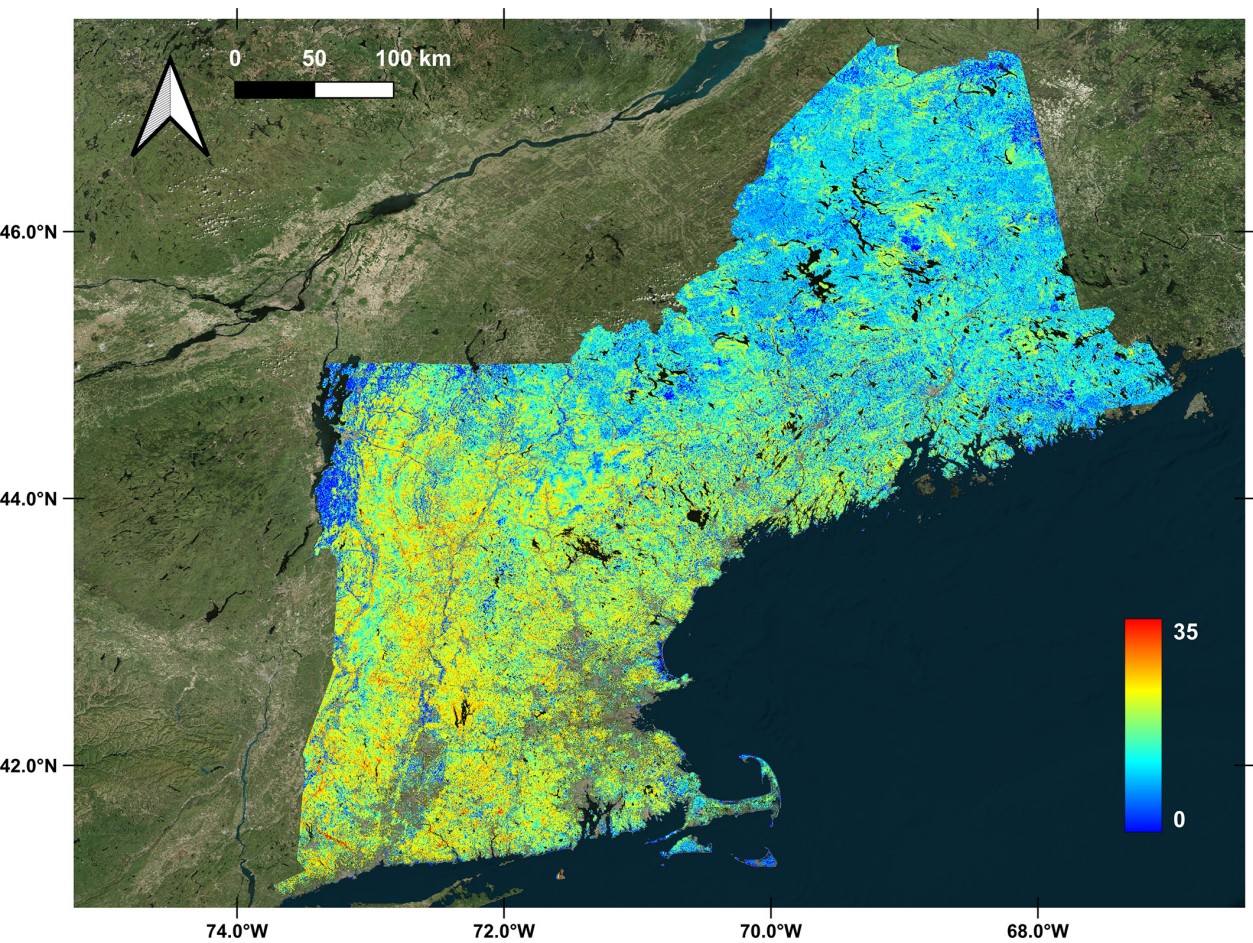

Figure 15: 30 m gridded forest height mosaic map based on ALOS-1 InSAR and GEDI RH98 metric for the New England region in the U.S., with a total area of 18million hectares. The color map ranges from 0 m ("blue" for bare surfaces) to 35 m ("red" for tall trees). It was projected onto the map coordinate of SRTM DEM products.

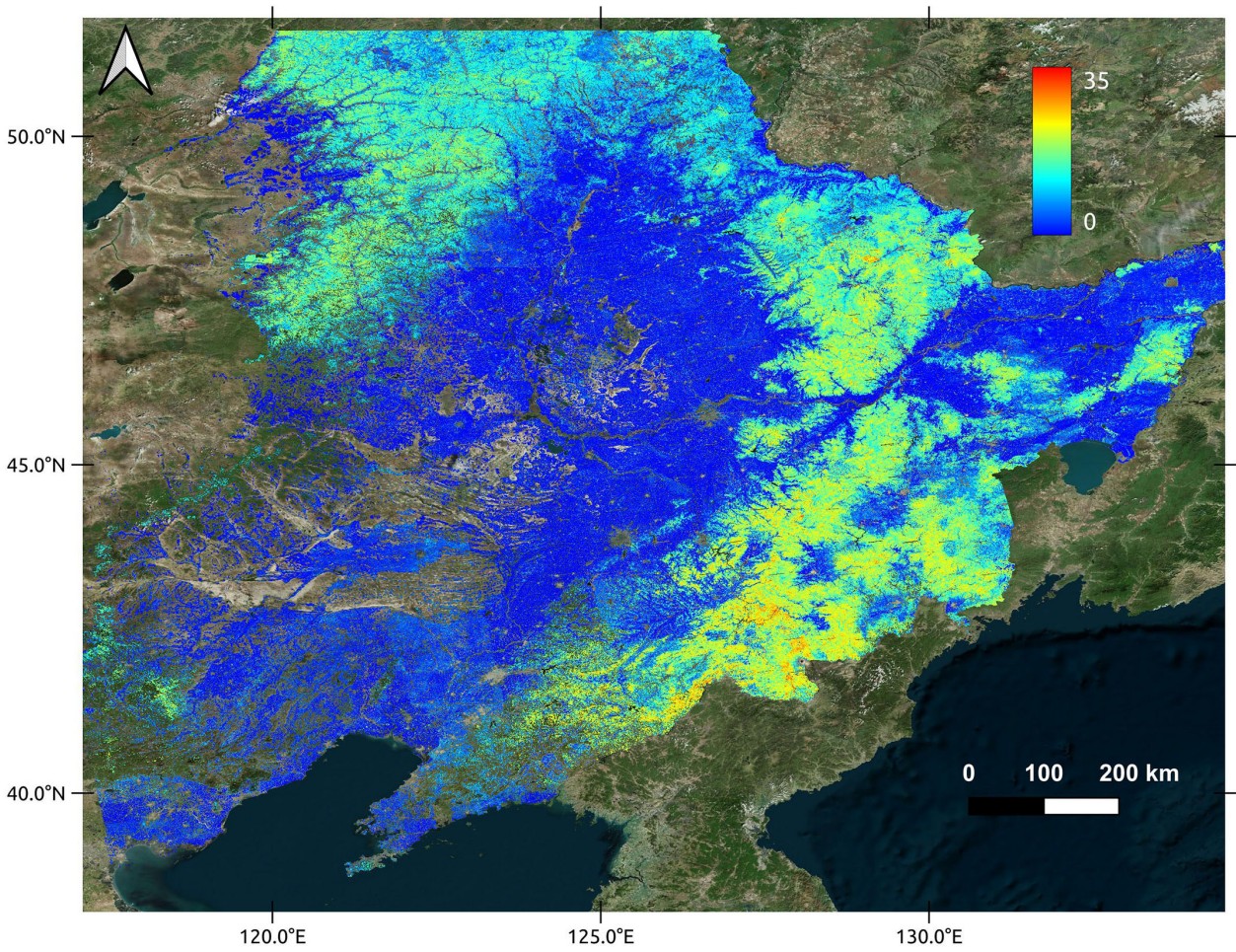

**Figure 16: 30 m gridded Forest height mosaic map based on ALOS-1 InSAR and GEDI RH98 metric for the northeastern region of China, with a total area of 152 million hectares.**

## 4.2 Validation over the New England region

This subsection presents the validation of forest height inversion across representative test sites in New England, U.S. First, we assess the accuracy of the inversion results for all selected forest sites using density scatterplots and corresponding error metrics including the Root Mean Square Error (RMSE), Coefficient of Determination (R2), Standard Deviation (STD), and Bias. Unless otherwise stated, all density scatterplots and associated error metrics in this study are based on a 0.8-hectare (3×3 pixels) aggregated pixel size. A comparative analysis is then conducted between our inversion results and two existing forest height products: 1) the 30-m resolution GLAD canopy height map, derived by fusing GEDI and Landsat time-series (Potapov et al., 2021), and 2) the 10-m resolution ETH canopy height map, generated by fusing GEDI and Sentinel-2 data (Lang et al., 2022). Both products are validated against reference LiDAR data.

Subsequently, case studies are presented to evaluate inversion performance under varying conditions. The Howland Forest site is analyzed to quantify improvements over our earlier methodology. A second case study focuses on the high-biomass region at the Harvard Forest site to assess inversion robustness in dense canopies. Finally, validation is extended to the White Mountain National Forest (WMNF) site with hilly topography, using high-resolution small-footprint LiDAR data as the validation data.

### 4.2.1 Summary of the validation results over the northeastern U.S.

Figure 17 presents density scatterplots comparing forest height estimates (derived from ALOS-1 mosaics or ALOS-2 single-scene inversions where suitable InSAR pairs are available) across all test sites in the New England region of the U.S. Error metrics for these estimates are reported in the corresponding scatterplots.

In short, the proposed inversion approach is capable of estimating forest height with an RMSE of 3-4 m in areas such as Howland and Harvard Forest sites, characterized by relatively flat topography and minimal human activity influence. In contrast, hilly or suburban areas like WMNF, GMNF, and Naugatuck State Forest (NSF) sites, exhibit slightly lower accuracy (RMSE 4-5 m). The ALOS-2 based inversion generally presents superior performance due to enhanced InSAR correlation behavior resulting from shorter temporal baselines, and less temporal discrepancy between radar and LiDAR data. For ALOS-1 inversion, the time mismatch between ALOS-1 and GEDI data is not a fatal problem as the inversion is carried out for temperate regions where the intact forest landscapes and forest height (of mature forests) remain stable. While our solution for addressing forest growth may not fully resolve temporal uncertainties, the resulting errors remain relatively minor, as evidenced by an RMSE of 3-4 m. This is also supported by the finding that the ALOS-1 based inversion is occasionally more accurate than ALOS-2 based estimates.

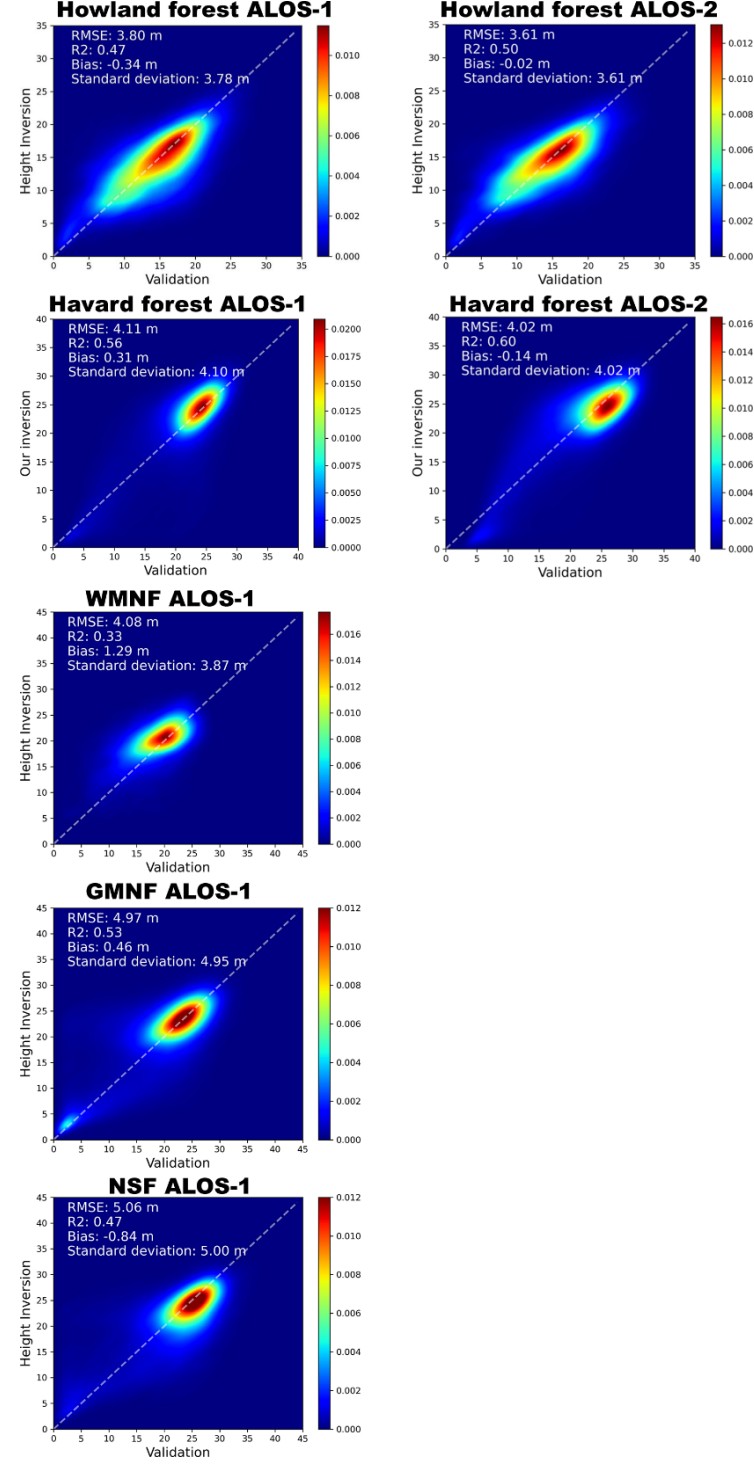

**Figure 17: Density scatterplots comparing LiDAR validation data with forest height inversion estimates across multiple sites of the New England: Left panels show ALOS-1-based estimates; right panels show ALOS-2-based estimates.**

In addition, we evaluate the inversion performance of two widely recognized global forest products: the GEDI-Sentinel (ETH) product (Lang et al., 2022) and the GEDI-Landsat (GLAD) product (Potapov et al., 2021). As shown in Table 4, both products exhibit significant biases across forest sites in the New England region compared with our ALOS-1 based estimates. Specifically, the GLAD product systematically underestimates canopy height, likely due to saturation effects in dense canopies, while the ETH product demonstrates larger systematic biases, consistently overestimating canopy height at all sites. These findings are consistent with the analysis by (Qi et al., 2025). In contrast, our inversion method achieves lower RMSE and smaller biases in most cases, with the exception of Naugatuck State Forest. At this site, hilly topography and suburban land cover likely contribute to reduced inversion accuracy. Notably, the ETH product exhibits lower STD and higher height-related $R^2$ values, likely attributable to its integration of high-resolution sentinel-2 data.

**Table 4 Comparison of GLAD, ETH, and our ALOS-1 based canopy height products with airborne LiDAR data across all forest sites in the New England region.**

| Validation sites | | GEDI-Sentinel (ETH) | GEDI-Landsat (GLAD) | GEDI-ALOS (Our product) |
|---|---|---|---|---|
| Howland Forest | RMSE | 5.71 | 5.59 | 3.81 |
| | R2 | 0.53 | 0.1 | 0.47 |
| | Bias | 4.6 | -1.87 | -0.34 |
| | Standard Deviation | 3.38 | 5.21 | 3.78 |
| Harvard Forest | RMSE | 4.79 | 5.66 | 4.11 |
| | R2 | 0.73 | 0.24 | 0.56 |
| | Bias | 3.35 | -0.78 | 0.31 |
| | Standard Deviation | 3.45 | 4.62 | 4.10 |
| White Mountain National Forest | RMSE | 5.98 | 5.46 | 4.08 |
| | R2 | 0.58 | 0.22 | 0.33 |
| | Bias | 4.34 | -0.87 | 1.29 |
| | Standard Deviation | 4.04 | 5.2 | 3.87 |
| Green Mountain National Forest | RMSE | 5.55 | 5.78 | 4.97 |
| | R2 | 0.69 | 0.18 | 0.53 |
| | Bias | 3.93 | -1.79 | 0.46 |
| | Standard Deviation | 3.92 | 5.09 | 4.95 |
| Naugatuck State Forest | RMSE | 4.89 | 5.07 | 5.06 |
| | R2 | 0.67 | 0.45 | 0.47 |
| | Bias | 2.83 | 1.16 | -0.84 |
| | Standard Deviation | 3.98 | 4.43 | 5.01 |

### 4.2.2 Howland Forest site

The Howland Forest was selected as one of the representative test sites, continuing from previous efforts in developing the
inversion approaches (Lei et al., 2018; Lei and Siqueira, 2022). Comparing results from these earlier studies with the current
work allows for an evaluation of performance improvements. A strip of LVIS LiDAR data acquired in 2009 was used as the
reference to assess the inversion accuracy of both the ALOS-1 based forest height mosaic map and the ALOS-2-based inversion
from a single pair of ALOS-2 data (frame: 890, orbit: 37). The comparison between the inverted forest height estimates and
the LVIS LiDAR data is presented in Figure 18, generating the differential height maps between the inversion and the LiDAR
data as shown in Figure 19 (a) and (b). For quantitative analysis, density scatterplots and corresponding statistical error metrics
comparing inversion results with validation data are displayed in the first row of Figure 17.

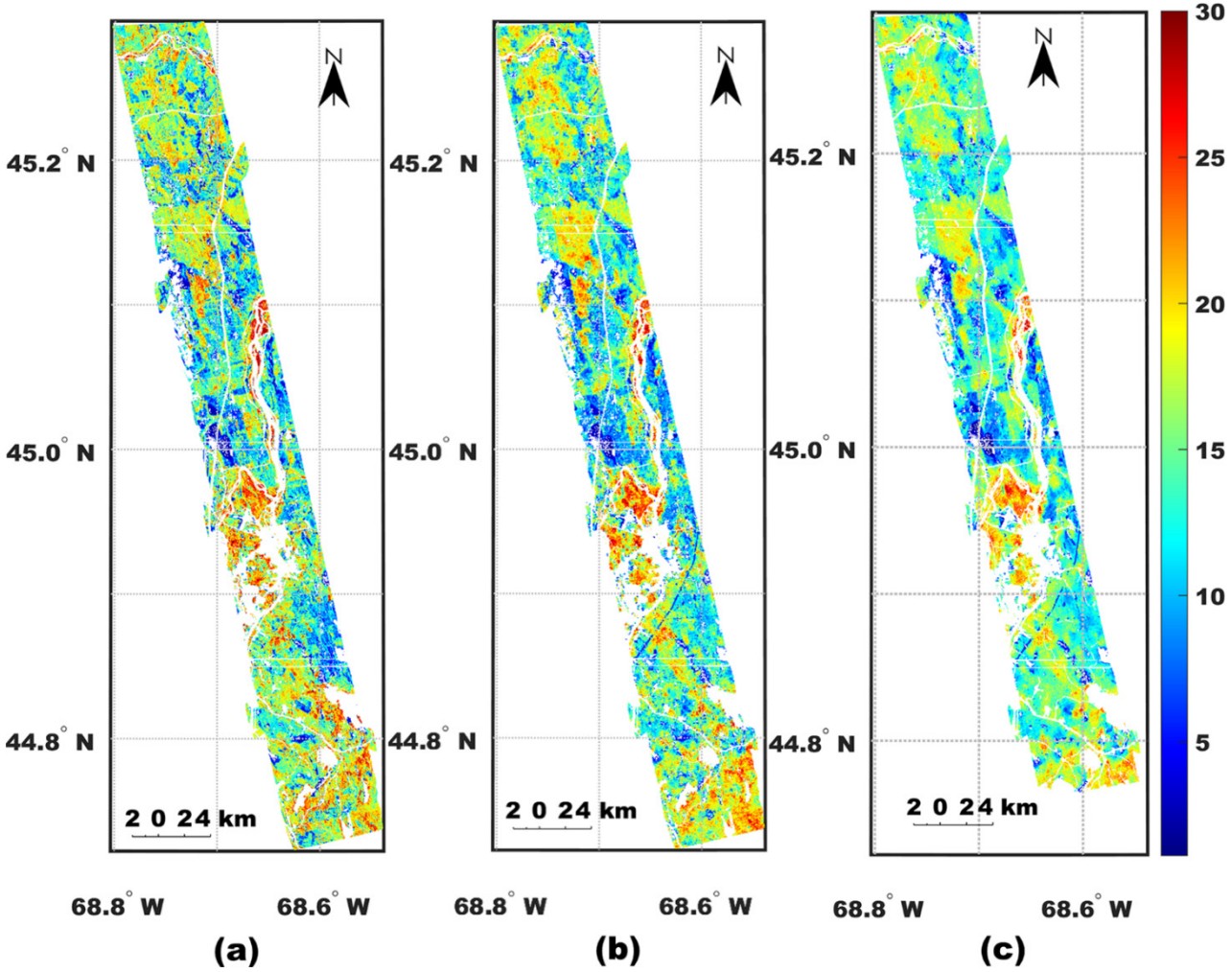

**Figure 18: Validation of forest height inversion results at the Howland Forest site: (a) LVIS RH98 canopy height map (30 m grid),**
**(b) inversion extracted from ALOS-1 mosaic, and (c) ALOS-2 based inversion.**

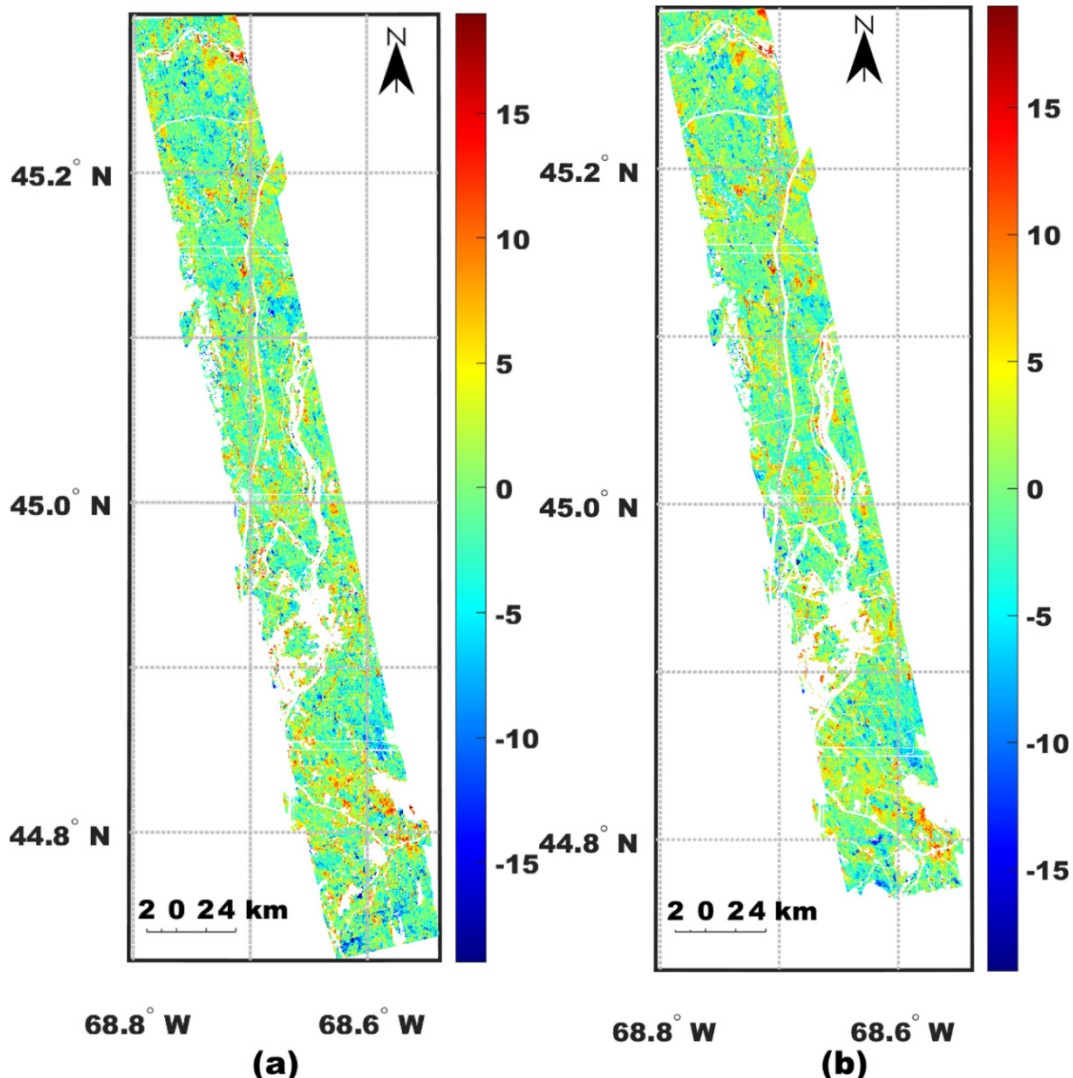

**Figure 19: The differential height maps of (a) ALOS-1 based inversion versus LVIS data and (b) ALOS-2 based inversion versus LVIS data.**

The comparison reveals that the ALOS-1-based mosaic inversion estimates forest height with an RMSE of 3.8 m. In contrast, the ALOS-2-based single-scene inversion achieves enhanced accuracy (RMSE = 3.6 m), likely due to improved correlation from its 14-day temporal baseline. To compare these results against our earlier work (Lei et al., 2019), we applied the global-fitting based inversion method (in the left panel of Figure 20), which yielded an inversion accuracy of RMSE = 4.38 m at an aggregated pixel size of 0.81 ha. This demonstrates that the global-to-local two-stage inversion approach significantly improves inversion accuracy, particularly over tall forest regions.

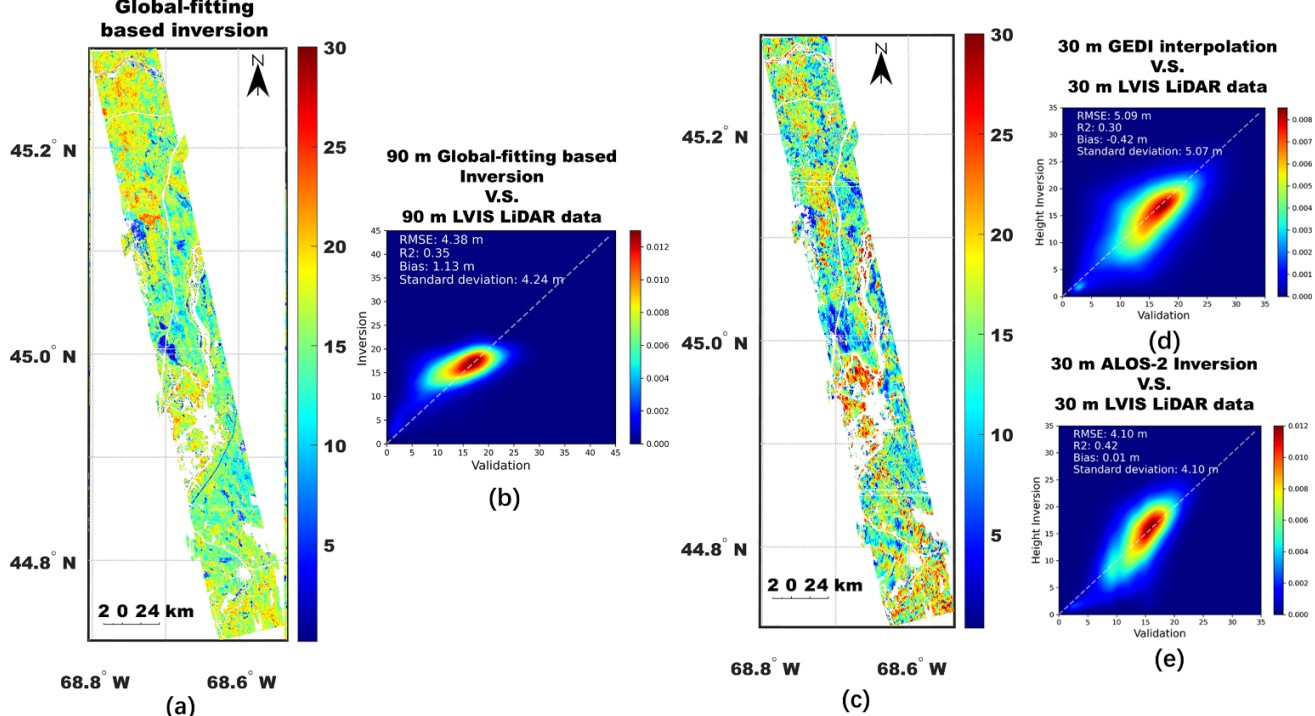

(a)

**Figure 20:** (a) global-fitting based inversion (Lei et al., 2019) applied to the scene ($S = 0.94, C = 9$), (b) comparing 90 m gridded maps from (a) with corresponding LVIS data; (c) interpolated 30 m gridded GEDI height map; (d) density scatterplot comparing the interpolated 30 m GEDI map with LVIS LiDAR data; (e) density scatterplot comparing ALOS-2-based inversion results with LVIS LiDAR data.


By synergistically combining SAR data with GEDI samples, this methodology not only resolves the wall-to-wall mapping limitations inherent to GEDI's discrete sampling but also improves inversion precision. As shown in the right panel of Figure 20, the 30 m interpolated GEDI based forest height maps still face discontinuity problems. However, an accuracy improvement of up to 20% has been achieved for ALOS-2 based estimates. This enhancement stems from the refined characterization of

temporal parameters $(S, C)$, enabled by leveraging GEDI's dense spatial sampling, as shown in Figure 21. As anticipated in Section 3.4.3, the saturation behavior of $S$ parameters arises from approximating the modified forest growth model $(15)$ using the original model $(3)$ in the local window with taller forests.

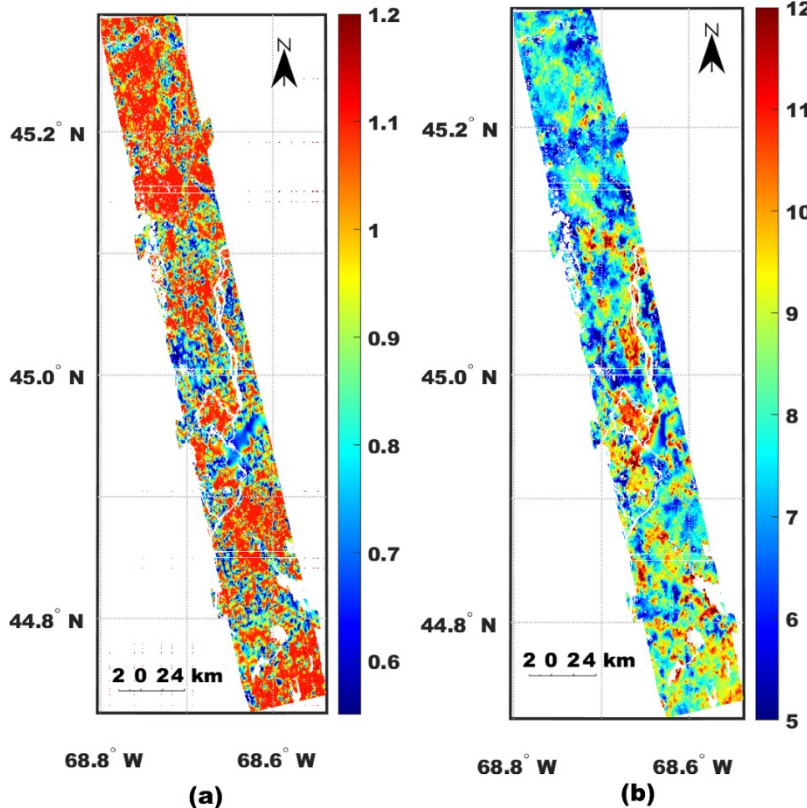

**Figure 21: the interpolated maps of temporal change parameters for (a) $S$ and (b) $C$**

### 4.2.3 Harvard Forest site

The Harvard Forest site was selected to evaluate the inversion in a region characterized by high biomass, up to 400 Mg/ha (Tang et al., 2021). Figure 22 presents the LVIS validation data acquired in 2021, covering the Harvard Forest area in subfigure (a), and the forest height estimates extracted from the ALOS-1 mosaic and the ALOS-2 single-scene inversion results (frame: 2770, orbit: 141) in subfigures (b) and (c). The comparison of ALOS-1 and ALOS-2 inversion results against the validation data is illustrated in the differential height maps in Figure 23. The density scatterplots from these two comparisons are given in the second row of Figure 17.

Both the ALOS-1 mosaic and ALOS-2 single-scene inversion are capable of estimating forest height with an RMSE of 4 m. The biased estimates occurred in taller forest stands may be attributed to the degraded sensitivity of GEDI measurements over dense forest stands (Fayad et al., 2022).

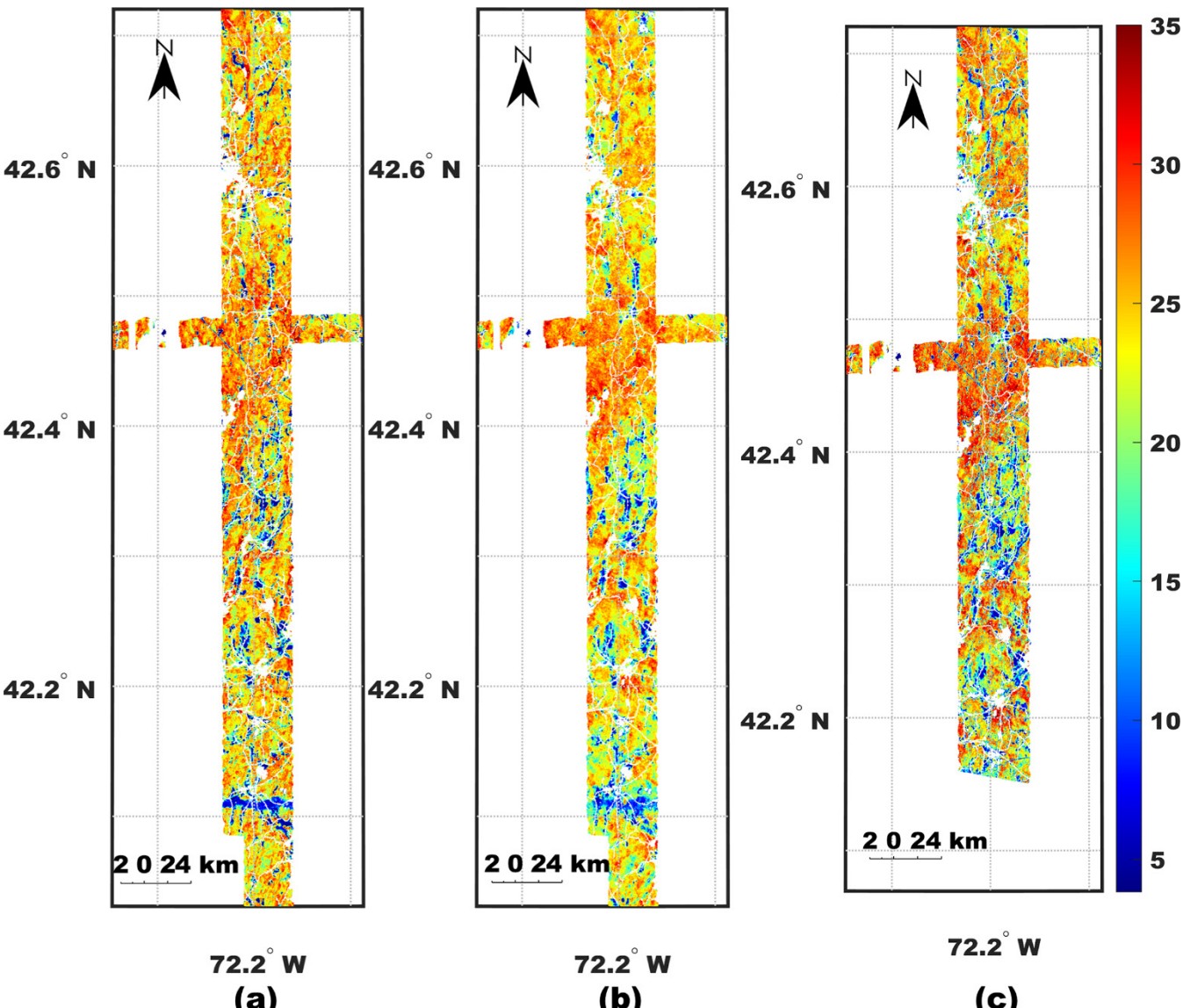

**Figure 22: Validation of 30 m gridded forest height inversion at the Harvard Forest site: (a) LVIS LiDAR RH98, (b) ALOS-1 based estimates (c) ALOS-2 based inversion.**

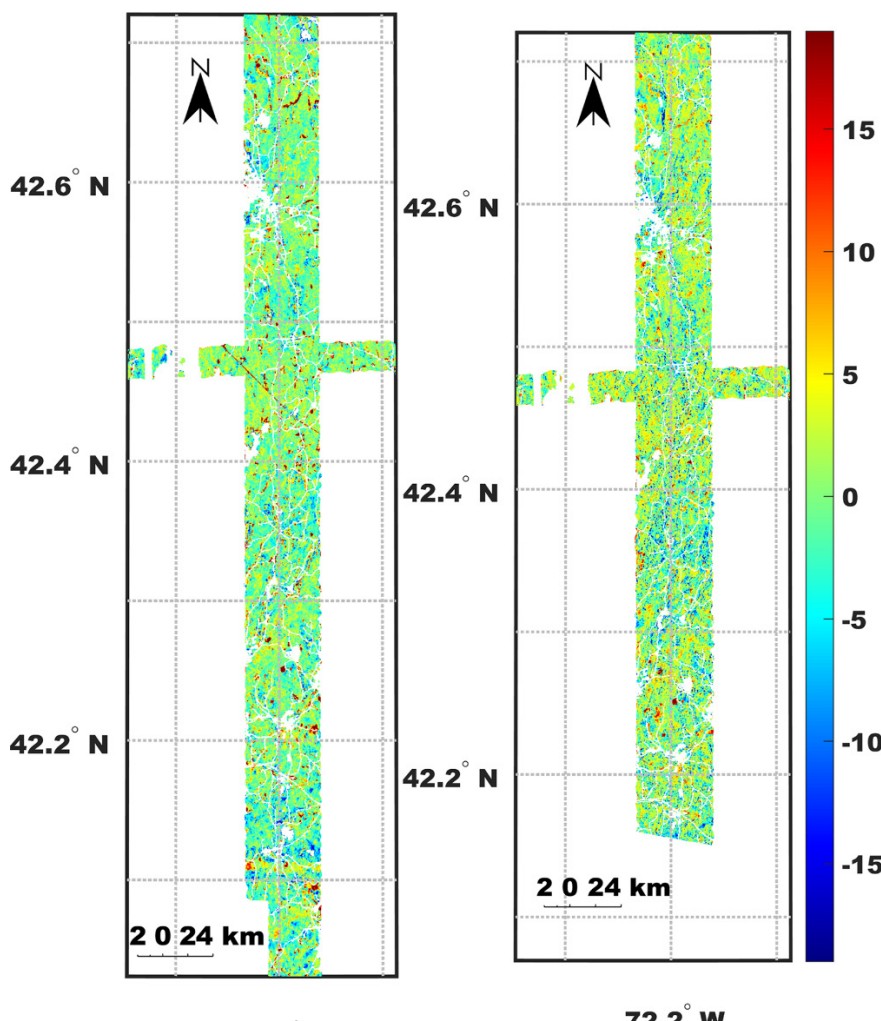

**Figure 23: Differential height maps over the Harvard Forest site: (a) ALOS-1 mosaic versus LVIS LiDAR, and (b) ALOS-2 single scene versus LVIS LiDAR.**


### 4.2.4 White Mountain National Forest site

The evaluation of the forest height inversion will also be extended to mountainous areas like the WMNF site, considering the potential challenges GEDI and InSAR observations might face in these regions. These challenges include GEDI's geolocation shifts and slope effects, as well as radar's viewing geometry problems (e.g., layover, shadow, foreshortening).


A high-resolution Canopy Height Model (CHM), derived from small-footprint GRANIT LiDAR data, serves as the validation reference. As outlined in Subsection 2.2.3, due to the footprint difference between small-footprint LiDAR data and GEDI

observations, an equivalent RH metric must be extracted within the same footprint size as GEDI observations to ensure comparability. Directly comparing GEDI-based forest height inversions with the reprojected CHM, via resampling or multi-pixel averaging, without this adjustment introduces significant bias. Figure 24 illustrates the difference between the ERH98 metric and the mean value (or ERH50 metric) in subfigure (a) and shows the maps of these two metrics in subfigures (b) and (c). Notably, ALOS-1-based forest height estimates (Figure 24 (d)) align closely with the ERH98 metric (Figure 24 (b)). The corresponding density scatterplot of the differential map is provided in the third row of Figure 17.

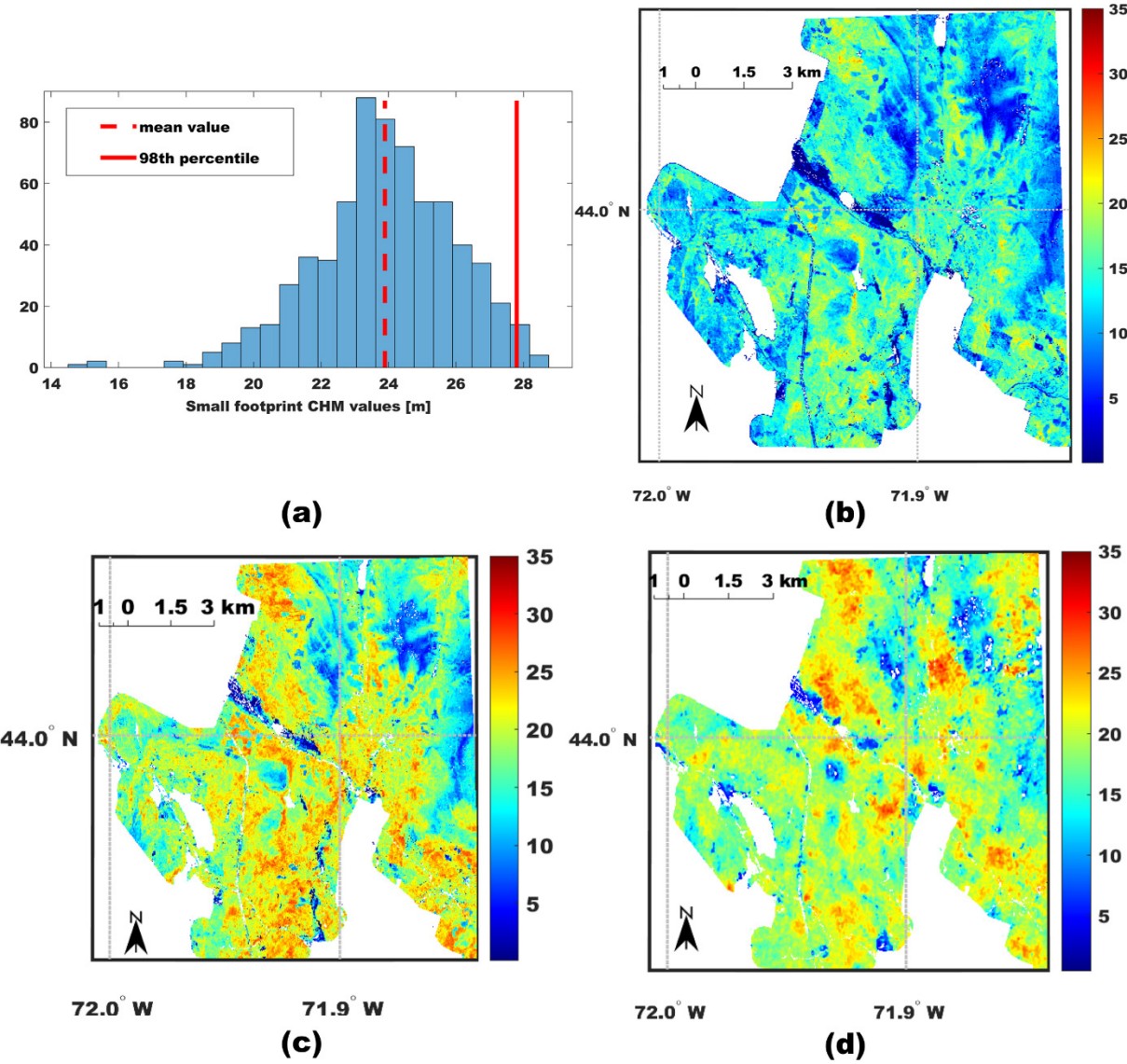

Figure 24: (a) An example of histogram formed by small footprint CHM values within the GEDI footprint, with mean height and a 98-th percentile height marked by the red dashed and red solid lines, respectively. (b) 30 m gridded reprocessed forest height based

on the ERH50 metric, (c) 30 m gridded reprocessed forest height based on the ERH98 metric and (d) corresponding forest height estimates extracted from ALOS-1 mosaic.

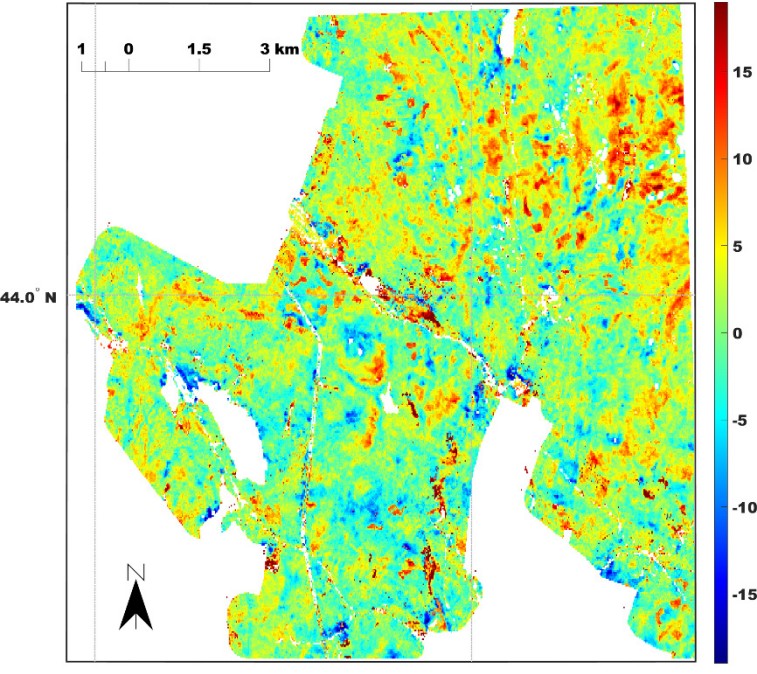


Figure 25: (a) Differential height map between the ALOS-1 mosaic and GRANIT ERH98 map and (b) the corresponding density scatterplots.

## 4.3 Validation against ALS data over the northeastern region of China

The forest height mosaic for northeastern China was validated exclusively against small-footprint airborne laser scanning (ALS) data at representative forest sites. These high-resolution ALS datasets were processed into ERH98 metric maps following the methodology detailed in Subsection 2.2.3. Initial analysis focused on density scatterplots across all surveyed sites. For a deeper investigation, two case studies were examined: Hubao National Park and Genhe Forest Bureau. Hubao National Park was selected due to the absence of significant forest disturbance, as demonstrated in Figure 4, while Genhe Forest Bureau was

chosen for testing the inversion performance over boreal forest bioregion in China.

### 4.3.1 Summary of the validation over the northeast of China

Forest height estimates across all sites across the northeast of China were validated against ERH98 metrics derived from small-footprint airborne LiDAR data. Density scatterplots as well as error metrics for the representative forest sites are summarized

in Figure 26. The highest accuracy was observed at the Mengjiagang Forest site, achieving an RMSE of 3.32 m and a R2 up

to 0.84. Most inversions exhibit a slight negative bias, likely due to GEDI's reduced signal penetration capability compared to airborne LiDAR. Slightly less accurate estimates are provided by the ALOS-2 based inversion at the Saihanba forest site, attributed to the limited overlapping area between the ALOS-2 single-scene inversion and ALS validation observations. This limitation arises from the distribution of heterogeneous land cover influenced by human activities. Overall, forest height estimates align closely with ERH98 LiDAR benchmarks, with accuracies of 3-4 m (even below 3.5 m at three sites) with R2 predominantly exceeding 0.65.

Notably, inversion performance in northeastern China surpasses results from GEDI calibration sites in the northeastern U.S., likely due to fewer forest disturbance in the selected represented forest sites (See Figure 4) compared to those in the northeastern U.S., as shown in Figure 3.

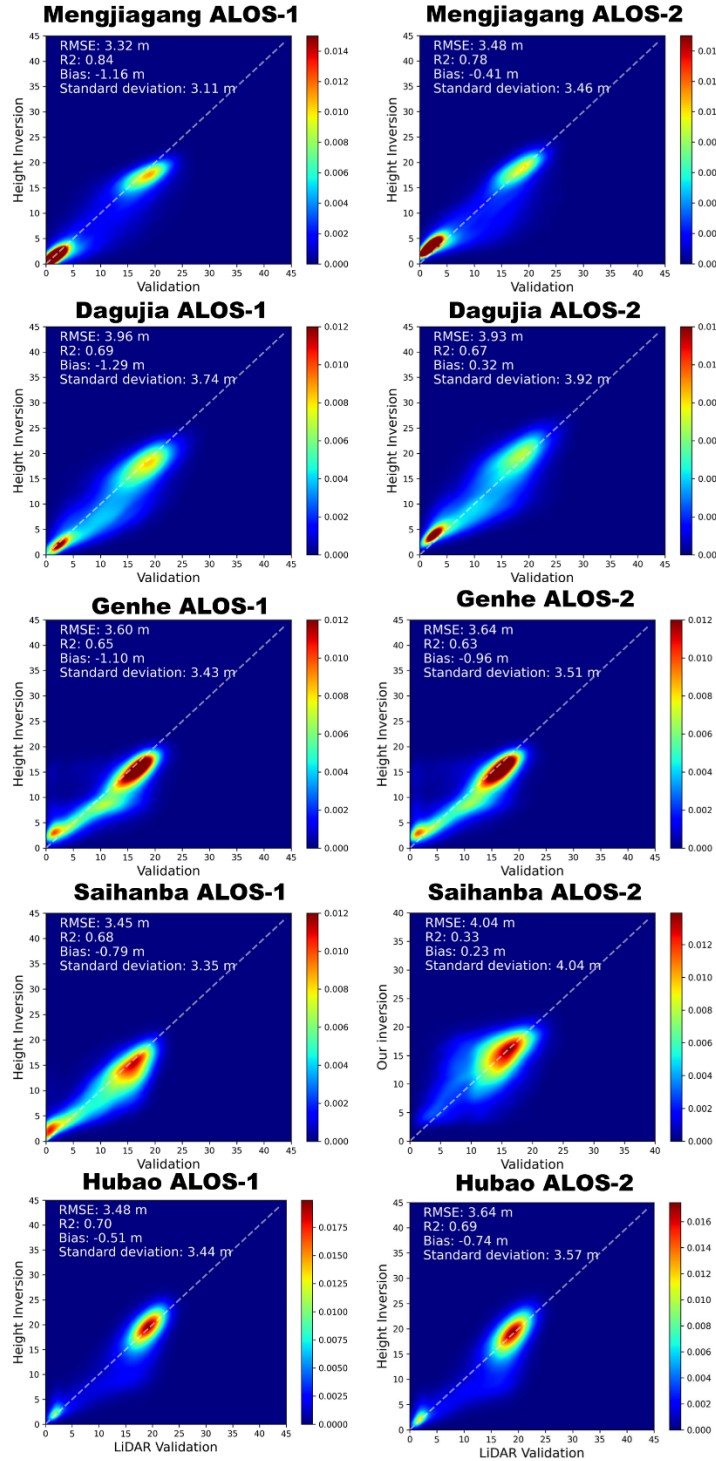

**Figure 26: Density scatterplots comparing LiDAR validation data with forest height inversion estimates across multiple sites of northeastern China based on the ALOS-1 InSAR observation (left panels), and the ALOS-2 observation (right panels).**

### 4.3.2 Hubao National Park Forest site

Hubao National Park is selected as it represents one of the typical temperate regions with the richest biodiversity in terms of wildlife and plants in the northern hemisphere. Figure 27: Panel (a) displays an ERH98 map derived from reprocessed 1 m resolution canopy height model (CHM) data acquired in 2018, using a window size same as GEDI footprints. Panels (b) and (c) present forest height estimates from ALOS-1 single-scene inversion (Frame 860, Orbit 421) and ALOS-2 single-pair InSAR data (Frame 860, Orbit 130), respectively. The detailed differential height maps are provided in Figure 28.

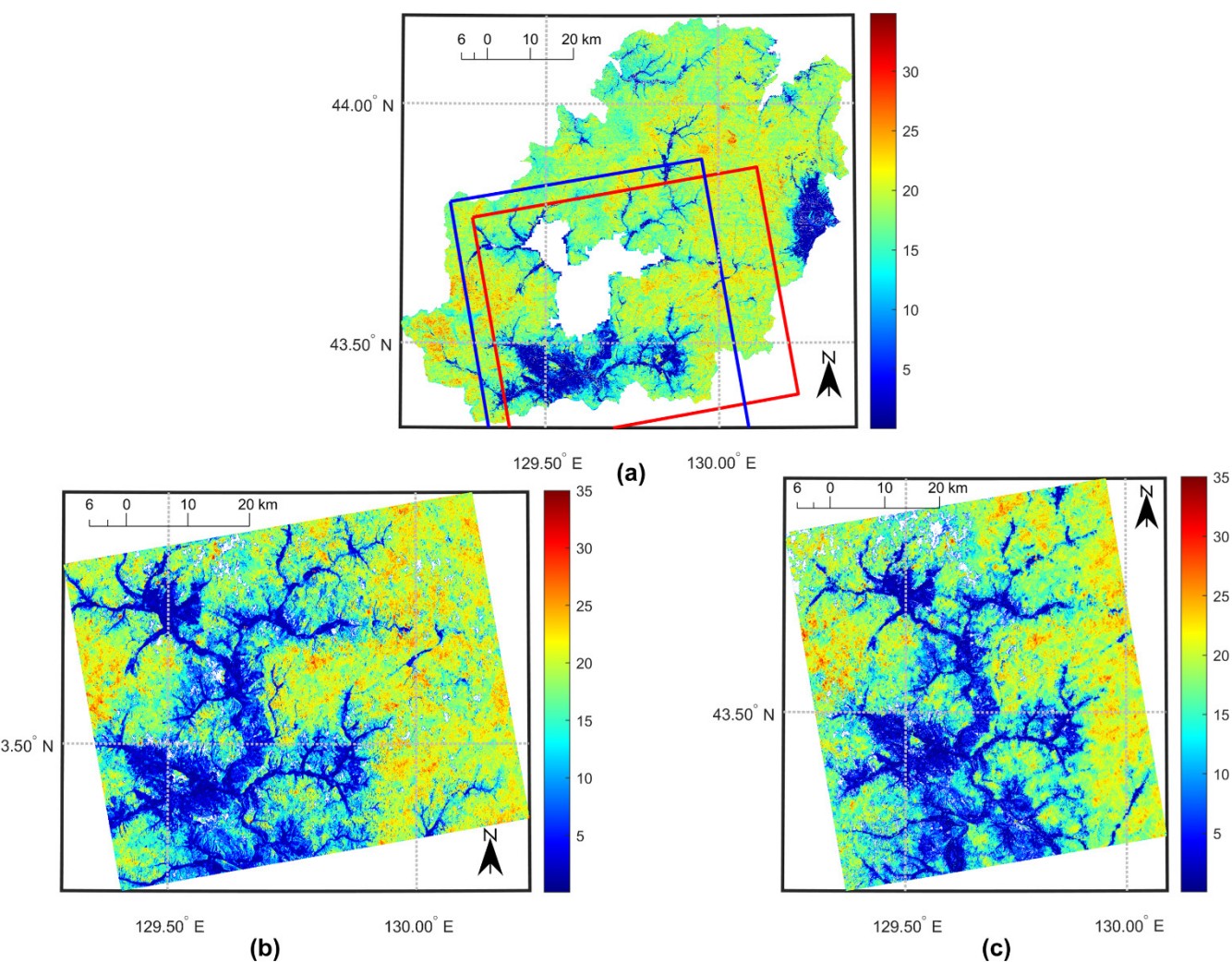

**Figure 27: Comparison of the forest height inversion with LiDAR data at the Hubao National Park site: (a) ALS ERH98 metric map. The red rectangle denotes the coverage of (b) ALOS-1 based single-scene inversion, whereas the blue rectangle indicates the coverage of (c) ALOS-2 based single-scene inversion.**

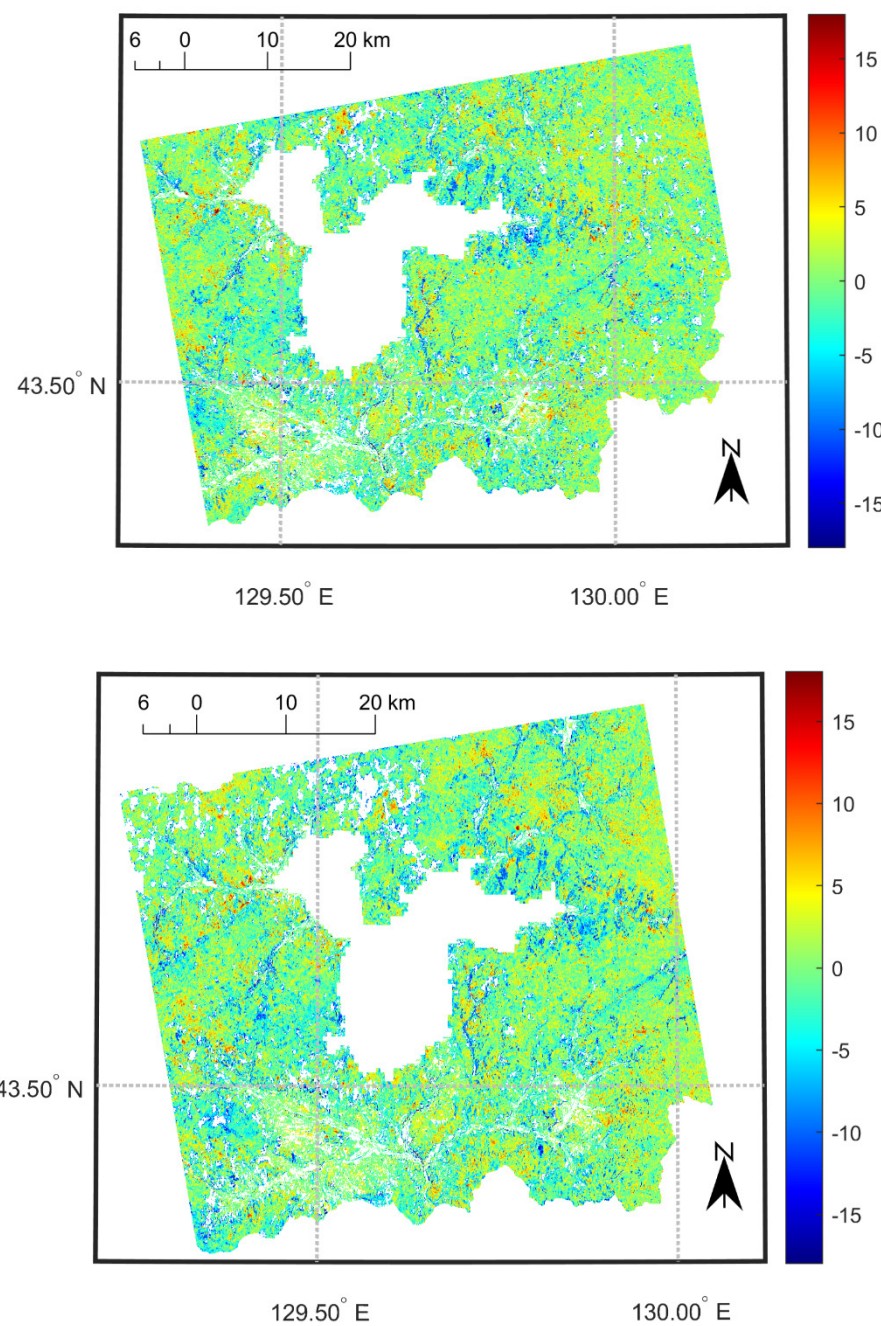

**Figure 28: Differential height map between (Upper panel) the ALOS-1 based inversion and ERH98 validation data, and between (Bottom panel) the ALOS-2 based inversion and ERH98 validation data.**

As shown in the density scatterplots (see the bottom row of Figure 26), both ALOS-1 mosaic and ALOS-2 single-scene inversions align closely with ALS derived ERH98 data, achieving accuracies of 3.5-3.8 m with a R2 up to 0.7. These results demonstrate the inversion precision comparable to airborne LiDAR measurements across both short and tall vegetation types. Differential height maps highlight discrepancies in transitional zones between forested and bare surfaces, underscoring the need for higher spatial resolution data integration (e.g., fusing TDX-GEDI data (Hu et al., 2024; Lei et al., 2021; Qi et al., 2025)) to refine estimates in such areas.

Slightly better performance for ALOS-1 based inversion is attributed to the fact that the ALOS-1 InSAR data archive offers the possibility to pick out the best InSAR pair with better correlation behavior; however, the availability of proper ALOS-2 InSAR data is limited.

### 4.3.3 Genhe Forest Bureau

A second case study examines the Genhe Forest Bureau, situated in northeastern China within one of the country's northernmost boreal forest regions. Figure 29 presents (a) the reference ERH98 map derived from 1 m resolution airborne LiDAR data, (b) forest height estimates from the ALOS-1-based mosaic, and (c) results from the ALOS-2 single-scene inversion. Differential height maps are shown in Figure 30. The spatial distribution of forest heights in both ALOS-1 and ALOS-2 inversions closely matches the reference ERH98 map, though primary discrepancies occur in short vegetation and bare surfaces, likely linked to anthropogenic disturbances.

Quantitative validation (see the third row of Figure 26) confirms strong agreement between inverted and reference forest height estimates, with an RMSE of 3.6 m and an R2 of 0.65, demonstrating the method's effectiveness and accuracy in boreal ecosystems.

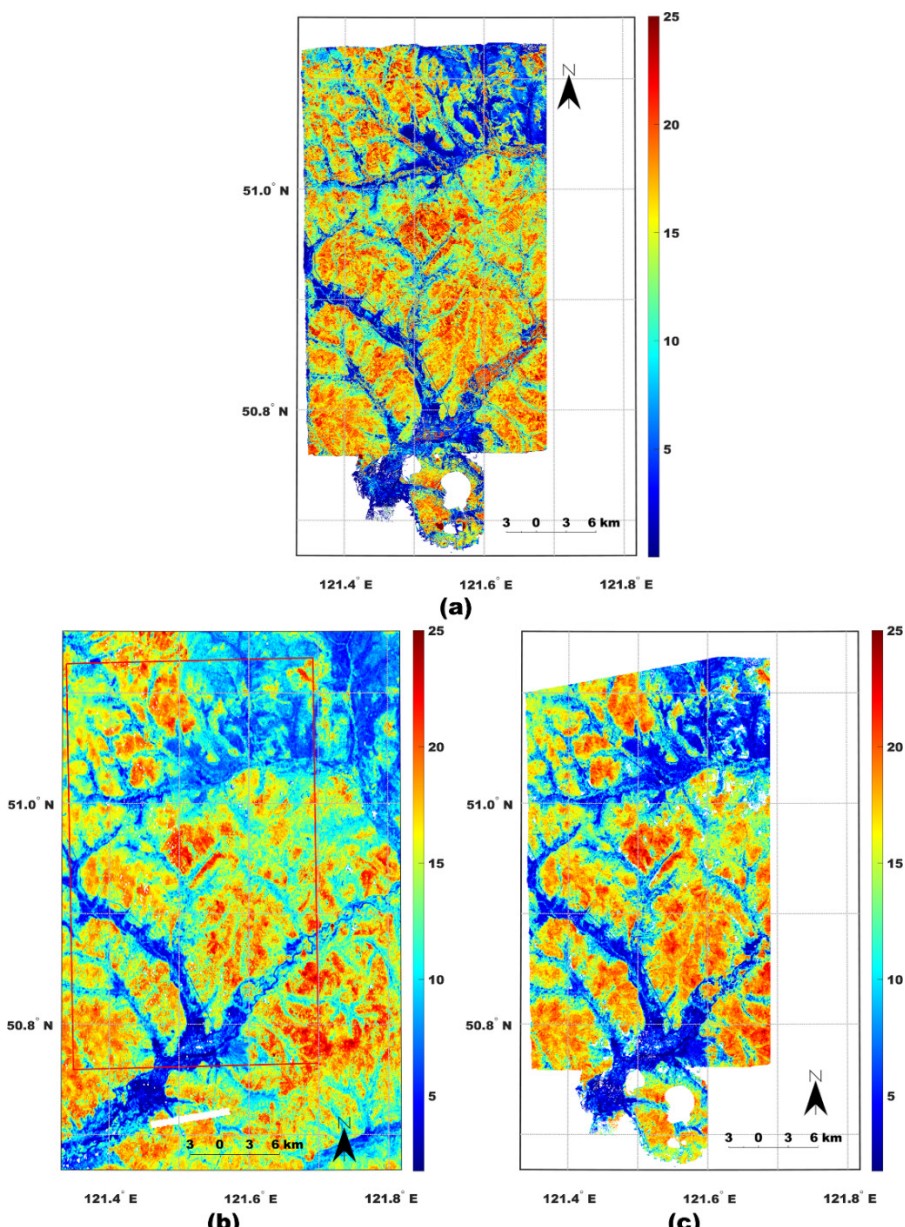

**Figure 29: Comparison of the forest height mapping over the Genhe Forest Bureau: (a) ALS ERH98 metric generated based on 25 m footprint; (b) ALOS-1 mosaic with red rectangle box indicating the overlapping area with (a), and (c) ALOS-2 based single-scene inversion.**

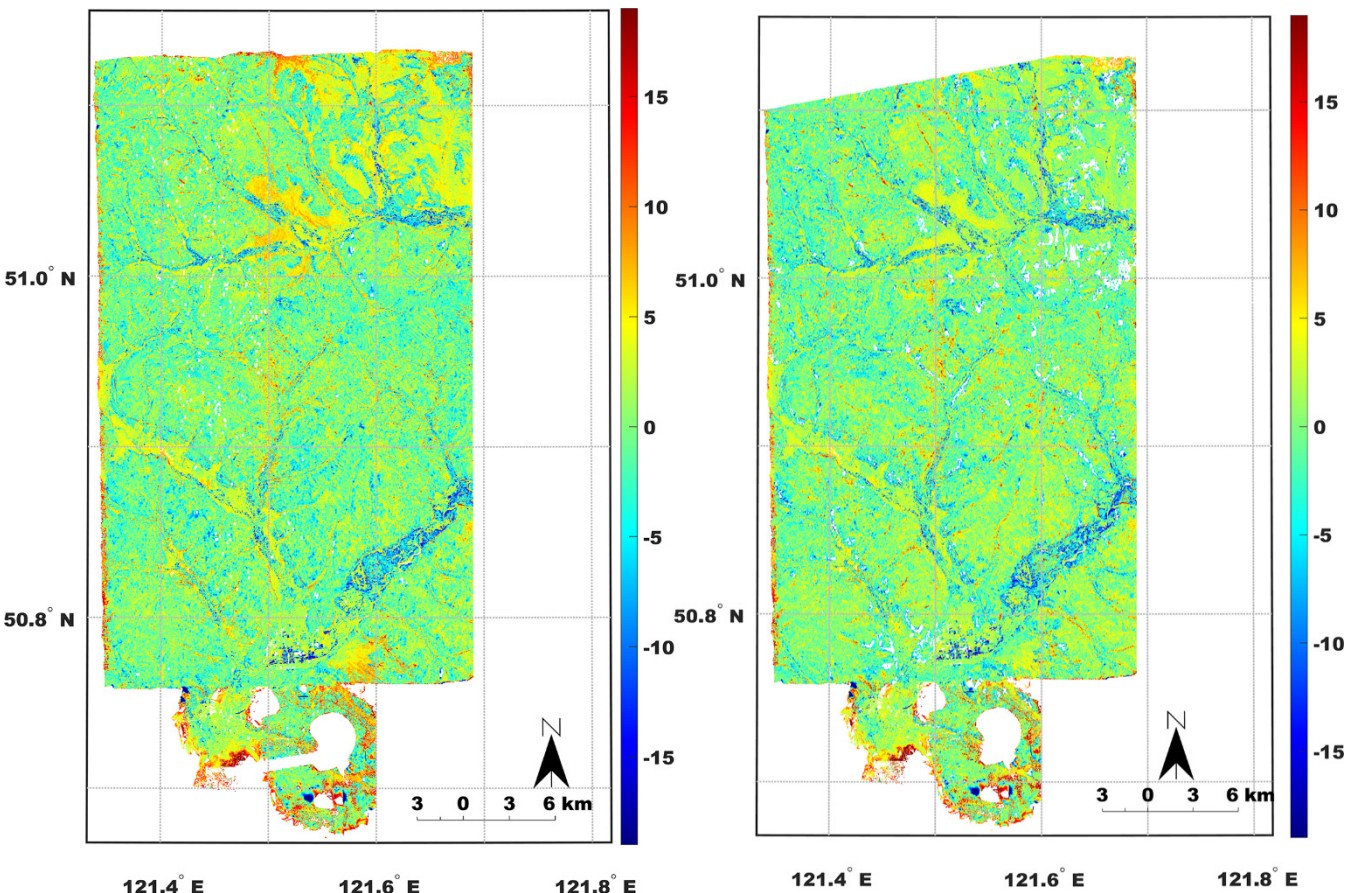

**Figure 30: Differential height map between (left panel) the ALOS-1 based inversion and ERH98 validation data, and between (right panel) the ALOS-2 based inversion and ERH98 validation data.**

## 5 Discussion

This section outlines key limitations of the proposed inversion inframework. A primary challenge stems from the complex InSAR correlation behavior induced by weather fluctuations (e.g., precipitation), particularly when constrained by a limited number of InSAR pairs. Future missions such as NISAR and BIOMASS (Quegan et al., 2019) are expected to mitigate this issue. As data stacks accumulate rapidly within each season, seasonally synthesized coherence maps could be produced by averaging or selecting maximum coherence values from all available pairs (Kellndorfer et al., 2022). For example, leveraging NISAR's 12-day repeat cycle would generate 7 interferometric pairs per season (or 30 annually), enabling the creation of seasonally or annually averaged coherence maps. These refined datasets could significantly enhance the robustness of forest height inversion.

To address the temporal mismatch between ALOS-1 and GEDI datasets, we propose a twofold solution to account for forest change dynamics. While this approach does not fully capture inherent forest variability, the achieved forest height estimation accuracy of 3–4 m/ha suggests that its impact is negligible in temperate regions, where intact forest landscapes exhibit minimal canopy height variation due to gradual tree growth.

The methodology also depends on precise, spatially representative calibration samples from GEDI. Two key challenges arise: (1) slope-induced biases in GEDI forest height estimates, and (2) sparse sampling coverage in boreal and equatorial tropical regions. The first limitation can be mitigated using the RH metric derived from slope-corrected waveforms (Wang et al., 2019). The second issue may be resolved by integrating complementary LiDAR datasets, such as NASA's ICESat-1/2 missions. Combining ALOS-1/2 data with both GEDI and ICESat-1/2 observations would improve the method's accuracy and
adaptability, particularly in tropical ecosystems.

    Finally, the current inversion framework employs a fixed-size, distance-based weighting window to perform local fitting. This approach could be enhanced by using an adaptively sptial-varying window size driven by multi-parameter classification.

**6 Conclusion**

This paper presented a global-to-local two-stage forest height inversion approach for large-scale forest stand height mapping using L-band spaceborne repeat-pass InSAR and spaceborne GEDI LiDAR. This work extended our previous efforts in forest stand height mapping (FSH: https://github.com/leiyangleon/FSH; Lei and Siqueira, 2014, Lei and Siqueira, 2015, Lei et al., 2018) at large scale by incorporating GEDI LiDAR samples for capturing local information. The sparsely yet extensively distributed LiDAR samples provided by the GEDI mission are used to parameterize the semi-empirical InSAR scattering
model and to obtain forest height estimates. Building on earlier works (Lei et al., 2019; Lei and Siqueira, 2022), this paper removed the assumptions previously imposed by the limited availability of calibration samples before, and developed a new inversion approach based on a global-to-local two-stage inversion scheme. An effective use of regional GEDI samples in this approach allows for finer characterization of temporal decorrelation patterns and thus higher accuracy in forest height inversion while also suppressing issues in individual GEDI sample, e.g., geolocation errors. This approach was supported by fusing the
ALOS-2 InSAR data and GEDI data under nearly concurrent conditions. This approach is further applied to open-access ALOS-1 data for testing its mapping capabilities at large scale. To address the temporal mismatch between ALOS-1 data and GEDI data, the introduction of fusing ALOS-2 backscatter data and GEDI data is able to detect disturbed forest areas. Furthermore, a modified signal model is derived for addressing natural forest growth over temperate forest regions where the intact forest landscape and forest height are stable with slight change. Without detailed forest growth data, simulations
confirmed that this modified model is found to be well-approximated with original model via local fitting based on simulation. For evaluating its performance, two forest height mosaic maps were generated to investigate the northeastern regions in the

U.S. and China, covering a total area of 18 million hectares and 152 million hectares, respectively. Validation of the forest height estimates demonstrates substantial accuracy improvements achieved by the proposed approach compared to the previous efforts, i.e., from 4 m RMSE for 6.25-hectare aggregated pixel size to 3.8 m RMSE for 0.81-hectare aggregated pixel size at the Howland Forest site. The proposed fusion approach not only addresses the sparse spatial sampling problem of the GEDI mission, but also improves the accuracy of forest height estimates compared to the GEDI interpolated height estimates by 20%. The extensive evaluation of forest height inversion against LVIS LiDAR data over northeastern U.S. indicates an accuracy of 3-4 m on the order of 0.81 hectare over smooth areas and 4-5 m over hilly areas, while the forest height estimates over northeastern China compare well with small footprint LiDAR validation data even at an accuracy of below 3.5 m on the order of sub-hectare and with R2 mostly above 0.6.

Despite the limitations outlined in Section 5, the method demonstrates promising forest height accuracy at the sub-hectare scale  using open-access InSAR and LiDAR datasets, underscoring its potential for integration with current and future missions (e.g., ALOS-4, NISAR, LuTan-1, MOLI, TECIS). This framework offers a cost-effective solution for large-scale forest height mapping based on open-acess remote sensing data, particularly in regions lacking multi-baseline bistatic InSAR data for advanced techniques like PolInSAR or TomoSAR.

## 7 Data availability

The forest height mosaics over the northeastern parts of U.S. and China are available at https://doi.org/10.5281/zenodo.11640299 (Yu and Lei, 2024). The used ALOS-1 data can be found via the Alaska Satellite Facility at https://search.asf.alaska.edu/#/?dataset=ALOS. GEDI data (from 2018 to 2023) can be downloaded from the EARTHDATA SEARCH website at https://search.earthdata.nasa.gov/. Regarding the validation data, the LVIS and GRANIT LiDAR data can be found at https://lvis.gsfc.nasa.gov/Data/Maps/GEDI2021Map.html and https://lidar.unh.edu/map/. Lei and Siqueira, 2022

## 8 Software Tools

The forest height mosaics are generated using the following software tools. First, ISCE with Version 2.4+ (https://github.com/isce-framework/isce2/releases; in particular the "stripmapApp" function) is used to preprocess the two ALOS-1/-2 images for procduing geocoded interferomteric coherence maps. Then, *FSH* Software Version 2 (https://github.com/Yanghai717/FSHv2) is used to invert and forest height by fusing GEDI and InSAR data and perform the mosaicking. Several preprocessing steps utilize basic Python libraries from FSH Version 1: https://github.com/leiyangleon/FSH.

## Acknowledgement

This study was sponsored by the National Key R&D Program of China (2022YFB3903300 and 2022YFB3903301) and the National Natural Science Foundation of China (Grant No. 62301529). The author would also like to thank the Japan Aerospace Exploration Agency for providing ALOS-1 and ALOS-2 data, and the National Aeronautics and Space Administration for providing GEDI data.

*Author contributions.* YL, PS, JC designed the study. YY, YL developed the processing software (*FSH v2*), YY,YL designed calibration/valalition tests and YY, XL, DG, AF,YP, WH carried out the validation tests. YY prepared the manuscript with contributions from all co-authors.

*Competing interests.* Authors declare that they have no conflict of interest.

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
