# Peer review of "Large-scale forest stand height mapping in the northeastern U.S. and China using L-band spaceborne repeat-pass InSAR and GEDI LiDAR data"

_Earth System Science Data, 2024_

## Author Response (AR1)

Authors' Response to Reviews of

Large-scale forest stand height mapping in the northeastern U.S. and China using L-band spaceborne repeat-pass InSAR and GEDI LiDAR data

Yanghai Yu, Yang Lei, Paul Siqueira, Xiaotong Liu, Denuo Gu, Anmin Fu, Yong Pang, Wenli Huang, Jiancheng Shi

**RC: Reviewers' Comments**, AR: Authors' Response,    ☐ Manuscript Text

**Reply to the first reviewer:**

**RC: Estimating forest height from InSAR and spaceborne lidar data over large areas are challenging but meaning work. However, the presentation of this manuscript makes it even more challenging to understand than it should be. Here are some comments that may be helpful!**

AR: Thanks very much to the reviewer for the recognition and valuable comments! The specific response and revision are listed below.

**RC: 1. Line 50: repeat "sensitive to".**

AR: Thanks for noting this typo. We have corrected this in the revised version at line 52, as shown below.

> …LiDAR and SAR are promising for capturing the internal vertical structure of forests: LiDAR is fundamentally sensitive to structural details, while radar detects the three-dimensional distribution of vegetation elements (Ulaby et al., 1990).…

**RC: 2. Line 70: use footprint instead of point. Point can be confused by lidar point cloud.**

AR: Thanks for this good suggestion. We have fixed it in the revised version at line 70:

> …However, GEDI collects only discrete footprint measurements, spaced approximately 60 meters apart in the along-track direction and 600 meters apart in the cross-track direction…

**RC: 3. Figure 2 and related text: Why not using Landsat/Sentinel based disturbance detection results, directly?**

AR: This is an insightful suggestion. The goal of this paper is to develop a self-contained inversion framework for large-scale forest height estimation by fully leveraging the Radar-LiDAR fusion. In our approach, the SAR backscatter-based forest height inversion (for heights below 10 m) replaces the InSAR coherence-based estimates. Furthermore, the openly accessible annual ALOS-2 backscatter mosaic products facilitate forest height estimation for short vegetation or bare surfaces after forest disturbance. In this way, land cover changes

from forests into short vegetation and bare surfaces will be included in the SAR backscatter-based forest height estimates. To demonstrate this approach, we tested how forest disturbances identified from Landsat/Sentinel-based disturbance products (used as ground truth benchmarks) emerges in the backscatter-based forest height pixels (<10 m). The experimental result showed that the SAR backscatter-based approach can effectively detect the majority of forest disturbances with comparable accuracy (>70%).

While optical-based products offer complementary insights, their direct integration could introduce additional artifacts or uncertainties, possibly in persistently cloud-covered regions. As the main scope of this paper is not to fully address the forest disturbance, we kindly ask for the reviewer's understanding that we do not introduce the optical-based products in current inversion framework.

**RC: Line 235 and equations: what does the a on the left mean? looks very similar to a. Suggest changing to other symbol. Also, where is hv(t2) ?**

AR: We feel sorry to create such confusion. This symbol is changed to the "$G$" to enhance the major distinction from symbol "a" at line 453 after revision. $h_v(t_2)$ represents the forest height after growth. However, the forest growth rate (although it can be derived by comparing the before/after status) can be modelled depending on the status of forest at either epoch, e.g., the initial forest height $h_v(t_1)$ in this work. Accordingly, this explanation is added at line 450 after revision as follows:

…where $h_v$ represents the time-dependent forest height, $t_1, t_2$ denote the initial and later epochs, respectively. Although the forest growth rate is derived by comparing pre- and post-growth states, it can be modelled based only on forest height data from either epoch, for example, using the initial forest height $h_v(t_1)$:

$$G = a \cdot h_v(t_1) + b \tag{12}$$

Where $a$ and $b$ are linear coefficients. If a dense time-series of forest height data over certain forest land cover is provided, the above equation can be constructed in a differential form as:…

**RC: Line 245: It seems to be a typical tree height based allometric equation. But the parameters would vary a lot among tree species, and the forest age, determined by both t1, and t2. How were these uncertainties addressed?**

AR: We fully agree with this insight. Forest growth rates are inherently species-dependent. However, obtaining species-specific growth rates for individual trees across large regions remains impractical.

To address this limitation, from a modeling perspective, we developed a modified model incorporating forest growth dynamics, simulating its behavior using statistical growth functions derived from spaceborne ICESat-1 and GEDI datasets. Simulations reveal that the modified signal model can be effectively approximated by

the original model at regional scales through local fitting, provided the adaptive parameters $S'$ and $C'$ are used to account for growth-induced variability (Figures 11 and 12 as shown below).

From a practical inversion perspective, we remark that the growth functions themselves were not directly employed in the inversion; instead, the original sinc model but with updated parameters $(S', C')$ is used in the actual application to approximate the modified model taking natural growth into account.

To clarify this, we have added statements at line 471 as follows:

…From a practical inversion perspective, the application of the modified model requires precise detailed statistics of natural growth across various forest types on a large scale. However, such data are currently unavailable, as existing spaceborne LiDAR datasets lack collocated measurements from two distinct time periods. In the absence of comprehensive forest growth data, this model is not yet recommended for direct large-scale use. Instead, it can be integrated into the framework of the original model by adjusting temporal parameters. The following subsection provides simulation examples to demonstrate this adaptation…

[Figure]

**Figure 11: Approximating the modified sinc model with the original model by aligning two points at 10 and 30 m: the newly fitted parameters at (a) Harvard Forest sites; (b) the forest site in Vermont; (c) Howland Forest site. The y-axis represents the coherence magnitude estimated from Equation (3) and (15).**

[Figure]

**Figure 12: Approximating the modified sinc model with the original model by aligning two points either at 10 and 15 m or at 25 and 30 m: the newly fitted parameters at (a) the Harvard Forest site; (b) the White Mountain National Forest site; (c) the Howland Forest site. The y-axis represents the coherence magnitude estimated from Equation (3) and (15).**

**RC: Line 250: What is the sinc model for? Whether I missed it, or it failed to be introduced clearly. But it seems to be a very important one. Not clear how the values in the y axis of Figure 4 were calculated?**

AR: Thank you for highlighting this oversight. The sinc function was omitted in the initial formulation and has now been added into the revised manuscript (Section 3.4.2, Line 459) as follows:

$$\left| \gamma_{t\&v}^{HV}(t_1) \right| = S(t_1) \cdot \text{sinc} \left( \frac{h_v(t_2) - b \cdot dt}{(1 - a \cdot dt) \cdot C(t_1)} \right) \tag{15}$$

where $\left| \gamma_{t\&v}^{HV}(t_1) \right|$ represents the InSAR coherence at initial time $t_1$, it follows the model is shifted and scaled with respect to the original model (3), $dt = t_2 - t_1$.

The y-axis values are calculated using Equations (3) and (15). We understand that this information is not clearly explained in the original text. To address this, we have added their explanations in the relevant figure captions for Figures 10, 11, and 12, as follows:

**Figure 10, 11, and 12: … The y-axis represents the coherence magnitude estimated from Equation (3) and (15).**

**RC: Figure 3: Again, as shown in Fig 3b, the growth rate varied a lot among site (species, age, and site condition). Also in Fig 3a, it should be a combined results of many different growth rates. These results further demonstrate it is unreasonable to apply a global model for the entire regions, even just for the New England region.**

AR: We appreciate this valuable insight. As clarified above, a single global forest growth model cannot adequately capture the forest growth dynamics across diverse bioregions. To address this, we developed a modified model to represent the natural forest growth process and compared its behavior with that of the original model in Figure 11 and 12 now. In terms of practical inversion, through local fitting procedures (as verified by simulation), we demonstrate that the original model framework can approximate the modified model with updated parameters. Therefore, the original inversion framework is actually used in the large-scale application without introducing growth functions.

**RC: 2.2.3: Oops, I got lost after the sinc model. Sorry.**

AR: We apologize for the confusion in the descriptions of the original and updated models. For clarification, we have added the sentences to illustrate the main objective of this subsection at line 490 as follows:

…Without detailed forest growth statistics, this subsection demonstrates the modified sinc model can be well approximated in the framework of the original sinc model but with updated parameters $(S', C')$ using simulation. This enables the large-scale application achieved in the framework of original model without detailed forest growth statistics…

**RC: Figure 9: The flowchart definitely should come first, as the Figure 1 or 2. Also, make it a more general and easy to understand for general readers.**

AR: Thank you for your constructive suggestion. In response to your feedback, we have repositioned the revised methodological flowchart to the beginning of Section 3 (now Figure 5), and modify it using more generalized description:

[Figure]

Figure 5: Block diagram of the workflow for generating forest height mosaic.

**RC: Figure 10: Labels on the color bar are too small to read. I would also suggest zoom into a few sub-figures of the study areas to show more details.**

AR: We thank the reviewer for this valuable suggestion. In response, we have updated the original figure to include a zoomed-in subsection with bolded labels and text to enhance clarity. This revised version is now presented as Figure 6 in the manuscript as follows:

[Figure]

**Figure 6 An illustrative example of the processing steps at the Howland Forest site: (a) the input ALOS-1 coherence magnitude map; (b) the GEDI rh98 samples; (c) the forest height estimates based on InSAR coherence information; (d) the backscatter based height estimates; (e) the final forest height map after replacing the estimates of short trees in (c) with the collocated pixels in (d); (f) is the airborne LVIS LiDAR data for validation.**

**RC: Section 3: I would suggest put these parts before the methods.**

AR: Thanks for the good suggestion! Section 3 (now renumbered as Section 2) has been placed ahead of the methodology section.

**RC: Table 2 and 3: Please add hom many plot or how large is the validate site in the table2 and 3.**

AR: Thanks for the good suggestion. The relevant information has been updated in the Tables 1 and 2 in the revised manuscript as follows.

**Table 1 The forest validation sites covered by the airborne LiDAR observation in the New England, U.S**

| Validation sites | Location | Dominated tree species | LiDAR data acquisition year | Slope statistics | ALS validation area (ha) |
|---|---|---|---|---|---|
| Howland Forest | 68°44′ W, 45°12′N | Red spruce (Picea rubens Sarg.) and eastern hemlock | 2009 | Mean: 2.3° STD: 5.3 | $4.77 \times 10^4$ |
| Harvard Forest | 72°11′W, 42°31′N | Red oak, Red maple, Black birch, White pine, Eastern hemlocc | 2021 | Mean: 5.5° STD: 4.6° | $4.87 \times 10^4$ |
| White Mountain National Forest | 71°18′W, 44°6′N | Red Spruce, Eastern Hemlock, American Beech, and Red Maple, | 2011 | Mean: 9.7° STD: 8.6° | $1.20 \times 10^4$ |
| Green Mountain National Forest | 73°04′W, 43°57′N | Sugar maple, American beech, red maple, yellow and paper birch | 2021 | Mean: 10.4° STD: 7.6° | $8.91 \times 10^4$ |
| Naugatuck State Forest | 73°00′W 41°27′N | Northern red oak, Mixed upland hardwoods, Yellow-poplar | 2021 | Mean: 5.2° STD: 4.5° | $3.58 \times 10^4$ |

**Table 2 The forest validation sites covered by the ALS validation data in northeastern China**

| Validation sites | Location | Dominated tree species | LiDAR data acquisition year | Slope statistics | ALS validation area (ha) |
|---|---|---|---|---|---|
| Mengjiagang Forest site | 130°42′E, 46°25′N | Coniferous plantations (Larix gmelinii and Pinus syvestris) | 2017 | Mean: 6.6° STD: 5.6° | $3.78 \times 10^4$ |
| Dagujia Forest site | 125°00′E, 43°21′N | Coniferous plantations (Larix kaempferi, Pinus koraiensis, etc) | 2018 | Mean: 13.5° STD: 7.4° | $3.66 \times 10^4$ |
| Saihanba Forest site | 117°18′E, 42°24′N | Larix principis-rupprechtii, Pinus syvestris, and Betula | 2018 | Mean: 8.7° STD: 7.3° | $2.98 \times 10^4$ |
| Genhe Forest bureau | 121°32′E, 50°47′N | Larix gmelinii, Betula platyphylla, Populus davidiana | 2022 | Mean: 7.0° STD: 5.4° | $1.09 \times 10^5$ |
| Hubao Forest site | 130°12′E, 43°28′N | Mongolian oak, Basswood, Betula platyphylla | 2018 | Mean: 8.7° STD: 7.4° | $3.99 \times 10^5$ |

**RC: Fig 17,19,21,22,24,27,29, and so on. These figures are just too small to read clearly not to mention compare them. A good comparison should map the difference between the estimated and ground truth (ALS ERH98? maybe in Figure 29).**

AR: We sincerely apologize for the inconvenience during the visual interpretation of previous figures. To address this, we have carefully revised all relevant figures with bolded labels and text to enhance readability and interpretation as shown in Figures. 17, 18, 19, 20, 21, 22, 23, 24, 25, 26, 27, 28, 29, and 30 after revision (included at the end of this response). Additionally, we have updated differential height maps (include Figures 19, 23, 25, 28, and 30) between inversion and airborne validation data in the corresponding forest sites.

**RC: Conclusions: I would suggest have a longer discussion and a short conclusion in separate sections.**

AR: Thanks for this suggestion. All the limitations and implications for future works are discussed in the Section 5. The Sections 6 briefly concludes this paper.

**Updated larger figures with bold fonts**

[Figure]

**Figure 17: Density scatterplots comparing LiDAR validation data with forest height inversion estimates across multiple sites of the New England: Left panels show ALOS-1-based estimates; right panels show ALOS-2-based estimates.**

[Figure]

**Figure 18 Validation of forest height inversion results at the Howland Forest site: (a) LVIS RH98 canopy height map (30 m grid), (b) inversion extracted from ALOS-1 mosaic, and (c) ALOS-2 based inversion.**

[Figure]

**Figure 19: The differential height maps of (a) ALOS-1 based inversion versus LVIS data and (b) ALOS-2 based inversion versus LVIS data.**

[Figure]

**Figure 20: (a) global-fitting based inversion (Lei et al., 2019) applied to the scene $(S = 0.94, C = 9)$, (b) comparing 90 m gridded maps from (a) with corresponding LVIS data; (c) interpolated 30 m gridded GEDI height map; (d) density scatterplot comparing the**

interpolated 30 m GEDI map with LVIS LiDAR data; (e) density scatterplot comparing ALOS-2-based inversion results with LVIS LiDAR data.

[Figure]

**Figure 21: the interpolated maps of temporal change parameters for (a) *S* and (b) *C***

[Figure]

**Figure 22: Validation of 30 m gridded forest height inversion at the Harvard Forest site: (a) LVIS LiDAR RH98, (b) ALOS-1 based estimates (c) ALOS-2 based inversion.**

[Figure]

**Figure 23: Differential height maps over the Harvard Forest site: (a) ALOS-1 mosaic versus LVIS LiDAR, and (b) ALOS-2 single scene versus LVIS LiDAR.**

[Figure]

**Figure 24: (a)** An example of histogram formed by small footprint CHM values within the GEDI footprint, with mean height and a 98-th percentile height marked by the red dashed and red solid lines, respectively. **(b)** 30 m gridded reprocessed forest height based on the ERH50 metric, **(c)** 30 m gridded reprocessed forest height based on the ERH98 metric and **(d)** corresponding forest height estimates extracted from ALOS-1 mosaic.

[Figure]

**Figure 25: (a) Differential height map between the ALOS-1 mosaic and GRANIT ERH98 map and (b) the corresponding density scatterplots.**

[Figure]

**Figure 26: Density scatterplots comparing LiDAR validation data with forest height inversion estimates across multiple sites of northeastern China based on the ALOS-1 InSAR observation (left panels), and the ALOS-2 observation (right panels).**

[Figure]

**Figure 27: Comparison of the forest height inversion with LiDAR data at the Hubao National Park site: (a) ALS ERH98 metric map. The red rectangle denotes the coverage of (b) ALOS-1 based single-scene inversion, whereas the blue rectangle indicates the coverage of (c) ALOS-2 based single-scene inversion.**

[Figure]

**Figure 28: Differential height map between (Upper panel) the ALOS-1 based inversion and ERH98 validation data, and between (Bottom panel) the ALOS-2 based inversion and ERH98 validation data.**

[Figure]

**Figure 29: Comparison of the forest height mapping over the Genhe Forest Bureau: (a) ALS ERH98 metric generated based on 25 m footprint; (b) ALOS-1 mosaic with red rectangle box indicating the overlapping area with (a), and (c) ALOS-2 based single-scene inversion.**

[Figure]

**Figure 30: Differential height map between (left panel) the ALOS-1 based inversion and ERH98 validation data, and between (right panel) the ALOS-2 based inversion and ERH98 validation data.**

**Reply to the second reviewer:**

RC: **This paper presents a Radar/LiDAR fusion approach that create large-scale mosaics of forest stand height. After carefully reviewing the manuscript, I found this work presents some interesting results. However, there are some concerns needed to be addressed and clarified for improving the manuscript.**

AR: *Thanks for insightful comments. These have greatly helped us to improve the quality of our manuscript. Specific revisions addressing your comments are outlined below.*

RC: **The method of this work is based on temporal decorrelation modeling, but the Introduction part only presents and cites the author's previous work, without mentioning other well-known temporal decorrelation models. The introduction needs to be further improved.**

AR: *Thank you for this constructive suggestion. We have added a dedicated paragraph in the Introduction Section (now at line 94) reviewing prior research on temporal decorrelation effect. The newly appended text is as follows:*

> …Temporal decorrelation has been a widely studied topic in InSAR research (Rocca, 2007; Ahmed et al., 2011; Bhogapurapu et al., 2024). (Zebker and Villasenor, 1992) proposed a Gaussian model to analyze oceanic scenarios, while (Monti-Guarnieri et al., 2020) summarized the signal models tailored for vegetated scenarios. (Askne et al., 1997) introduced a coordinate-dependence of the vertical motion profile to analyse InSAR temporal decorrelation effects caused by wind. Building upon the well-known RVoG model, several signal models have been developed to explicitly incorporate temporal decorrelation effects (Lavalle et al., 2012; Papathanassiou and Cloude, 2003; Lei et al., 2017a).…

RC: **How is the size of the local window determined? The spatial density of GEDI is uneven, so why not use an adaptively varying window? Moreover, the local modeling approach is very similar to the work by Hu et al. (https://doi.org/10.3390/rs16071155), and they used an automatically varying window. It is recommended to add a citation to help readers better understand. In addition, using distance-based weighting does not seem to align well with the rapidly changed forest scenario. This should be further described and discussed.**

AR: *Thanks for these critical comments. The work by Hu et al., 2024 has been added in the literature review at line 88 as follows:*

> …, while (Hu et al., 2024) exploited local ICESat-2 LiDAR information, using regional polynomials and an adaptive window, to estimate equivalent forest phase centers under homogeneous forest and terrain conditions…

The detailed responses to the specific comments are listed as follows:

**The choice of window size and why not use an adaptively spatial-varying window?**

The optimal local window size is determined by minimizing the root mean square error (RMSE) of forest height estimates for moderate-to-tall trees (>10 m) across diverse window size, validated against independent lidar datasets. Empirical analysis identified a 960 m window as the optimal configuration, balancing spatial resolution, statistical robustness, and computational efficiency. This part of information is added at line 398 as follows:

> … The optimal local window size is determined by minimizing the root mean square error (RMSE) of forest height estimates for moderate-to-tall trees (>10 m) across diverse window size configurations, validated against independent lidar datasets. The selection of window size is a compromise between smooth and detailed information. This window size is selected here for including enough samples for model fitting while maintaining local detailed information…

The window size can be adjusted as a spatially varying parameter, provided that validation data are available. We admit an adaptively spatial-varying window could improve the modeling accuracy. At present, it is highly scene-dependent and could lead to unpredictable behavior in large-scale inversion scenarios. For example, very homogeneous scenario within smaller window size may induce ill-conditioning in the inversion system, a scenario rigorously avoided in our large-scale implementation. Additionally, employing variable window sizes would reduce computational efficiency, which is particularly undesirable in GPU based parallel computing architecture. Therefore, we prefer a fixed larger window to incorporate sufficient forest height variability and ensure a simplicity, computational efficiency, and robustness over large-scale application.

However, we believe integrating a more sophisticated classification-driven adaptive window strategy in the future will improve our local modeling accuracy. So, we include this consideration at line 822 in the Discussion Section as follows:

> … Finally, the current inversion framework employs a fixed-size, distance-based weighting window to perform local fitting. This approach could be enhanced by using an adaptively spatial-varying window size driven by multi-parameter classification.…

**The difference between our work and Hu et al., 2024:**

We agree that the local concept presented in this study shares some similarities with the work by Hu et al., 2024.

This study advances the global-to-local inversion framework initially proposed in our earlier work (Lei and Siqueira, 2022; Yu et al., 2023), extending it for large-scale applications. By integrating local GEDI lidar

samples, we calibrate a semi-empirical and semi-physical repeat-pass InSAR model at finer spatial scale, enabling improved forest height inversion accuracy. The framework's primary objective is to achieve large-scale, high-resolution forest height mapping using open-access SAR and LiDAR datasets. A core assumption of this method is that temporal decorrelation model remains spatially invariant at regional scales, while allowing for variability in forest height observations within these regions, while permitting variability in forest height observations across these areas, i.e., a large window of inhomogeneous forest (but with uniform temporal change parameters) is preferred.

In contrast, the work by Hu et al., 2024 used the local information of ICESAT -2samples around each pixel to build a polynomial model (3 order) for estimating the scattering phase center. The polynomial fitting is used to establishing the relationship between scattering phase center and InSAR coherence as well as slopes within the regional scale. Furthermore, they claim that the similar forests and similar slope conditions should be privileged to enable better estimates. The basic assumption is that the forest height within the local window is similar, so they can gain benefits with more homogenous window, i.e. a small window of homogeneous forest is preferred.

To clarify the distinction between our approach and that of Hu et al. (2024), we have added the following statements at line 112:

> …By efficiently leveraging regional GEDI samples, this approach calibrates a semi-empirical, semi-physical repeat-pass InSAR model at a finer spatial scale, substantially improving forest height inversion accuracy. The method assumes that the temporal decorrelation model remains spatially invariant at the regional scale, while permitting variability in forest height observations within those regions.…

And the main contribution of this paper is stated at line 124 as follows:

> …The key contribution of this paper lies in the use  local GEDI information for Radar-LiDAR data fusion, enabling large-scale and efficient forest height mapping using open-access spaceborne data, such as GEDI and forthcoming NISAR (Siqueira et al., 2024; Kellogg et al., 2020) data…

**For rapidly changed forest scenario,**

Sorry that we are not fully sure which specific scenario the reviewer is referring to: spatially rapid change of forests or temporally rapid change of forests. To ensure thoroughness, we will reply to both cases in response to the reviewer's comments.

**For a spatially-varying condition**, the current inversion framework employs a distance-based weighting window, primarily to ensure computational efficiency and avoid ill-conditioned scenarios as clarified above. For highly spatially heterogeneous cases, one potential solution involves leveraging classification algorithms, driven by external data sources, to prioritize targets sharing specific features. This part has been added to the Discussion Section. Alternatively, using high-resolution datasets (e.g., TanDEM-X) could better resolve spatial variations. We emphasize this advantage in our literature review, at line 81 as follows.

> …Without temporal decorrelation effects, TanDEM-X data offer opportunities to leverage very-high-resolution observations for addressing spatially heterogeneous landscapes…

**For a temporally-varying condition**, we integrate time-coincident ALOS-2 backscatter data to estimate heights of regenerating or low-stature vegetation in affected areas to mitigate the impact of rapid forest disturbances (e.g., logging or wildfires). However, our current framework is optimized for stable or slowly evolving temperate and boreal forests and cannot robustly resolve highly dynamic tropical forest changes, which lie beyond the scope of this study. As noted in the Discussion (Section 5), future missions such as NISAR or BIOMASS, providing dense time-series InSAR datasets, could enhance the temporal sampling required to capture rapid height changes in tropical ecosystems.

**References:**

Hu, H., Zhu, J., Fu, H., Liu, Z., Xie, Y. and Liu, K., 2024. Automated estimation of sub-canopy topography combined with single-baseline single-polarization TanDEM-X InSAR and ICESat-2 data. *Remote Sensing*, *16*(7), p.1155.

Y. Lei and P. Siqueira, "Refined Forest Stand Height Inversion Approach with Spaceborne Repeat-Pass L-Band SAR Interferometry and GEDI Lidar Data," *IGARSS 2022 - 2022 IEEE International Geoscience and Remote Sensing Symposium*, Kuala Lumpur, Malaysia, 2022, pp. 6388-6391, doi: 10.1109/IGARSS46834.2022.9884755.

Y. Yu, Y. Lei and P. Siqueira, "Large-Scale Forest Height Mapping in the Northeastern U.S. using L-Band Spaceborne Repeat-Pass SAR Interferometry and GEDI LiDAR Data," *IGARSS 2023 - 2023 IEEE International Geoscience and Remote Sensing Symposium*, Pasadena, CA, USA, 2023, pp. 1760-1763, doi: 10.1109/IGARSS52108.2023.10281488.

**RC: The author emphasizes using the backscatter coefficient to estimate low forests, but there is no physical explanation for how the 10-m threshold is determined. Additionally, forest height below 10m can undergo significant changes, such as in young forests and shrub. When using ALOS data with a significant time difference for inversion, how is the height variation of short forests taken into account?**

AR: Thank you for your constructive feedback. The definition of "low forest" (forests with canopy heights below 10 meters) aligns with established literature on shrubs, typically under 6–10 m (20–33 ft) in height (Lawrence, 2013; Allaby, 2012). These references have been added to the line 309 as follows.

> …which is then used to obtain backscatter-to-height estimates (Lei et al., 2019). Short trees are identified using a criterion where backscatter-derived forest height estimates fall below 10 meters, based on the maximum height of shrub (Lawrence, 2013; Allaby, 2012), and empirical studies (Lei et al., 2019)…

We fully recognize that forest stands below 10 meters in height undergo dynamic and rapid changes. To address this, we utilized recent ALOS-2 backscatter mosaic data (2019–2020), selected for its close temporal alignment with GEDI measurements, to estimate the height of short vegetation.

For those short forests experiencing significant temporal height variation within a short interval, the current ALOS-1/2 datasets lack the temporal resolution to resolve this. These can be more effectively resolved through dense time-series data from time-series TanDEM-X, Sentinel-1 data and forthcoming NISAR data. To clarify this point, we added this discussion at lines 436-438 of the manuscript as follows:

> …Short forests undergoing rapid temporal height variations within short intervals cannot be adequately captured by current ALOS-1/2 datasets. These dynamic changes can be better resolved using dense time-series data from TanDEM-X (Treuhaft et al., 2017; Lei et al., 2018), Sentinel-1 (Bhogapurapu et al., 2024), and the forthcoming NISAR mission…

**References:**

Lawrence, A.: Plant identification: creating user-friendly field guides for biodiversity management, Routledge2013.
Allaby, M.: A dictionary of plant sciences, Oxford University Press2012.
Treuhaft, R., Lei, Y., Gonçalves, F., Keller, M., Santos, J. R. d., Neumann, M., and Almeida, A.: Tropical-Forest Structure and Biomass Dynamics from TanDEM-X Radar Interferometry, Forests, 8, 277, 2017.
Lei, Y., Treuhaft, R., Keller, M., dos-Santos, M., Gonçalves, F., and Neumann, M.: Quantification of selective logging in tropical forest with spaceborne SAR interferometry, Remote Sensing of Environment, 211, 167-183, https://doi.org/10.1016/j.rse.2018.04.009, 2018.
Bhogapurapu, N., Siqueira, P., and Armston, J.: A new InSAR temporal decorrelation model for seasonal vegetation change with dense time-series data, IEEE Geoscience and Remote Sensing Letters, 2024.

**RC: The content and structure of Section 4 are very redundant, with many figures and tables conveying the same information. Figures 26 and 31 are completely redundant; they have already appeared earlier in the manuscript, so why show them again? Additionally, the content of Tables 4 and 5 is the same as what is shown in the figures 26 and 31? Also, note that the accuracy metrics in**

**Figure 26(b) are different from those in Table 4, please make the correction. In summary, the section 4 needs major adjustments and improvements.**

AR: We agree that there was redundant content, and have revised the text accordingly.

To enhance clarity and reduce redundancy, we have restructured Section 4 as follows: all the density scatterplots for the New England and northeastern China study areas are now firstly summarized in Figure 17 and 26 in Subsections 4.2.1 and 4.3.1, respectively. This reorganization allows these scatterplots to serve as the references for subsequent detailed analyses of representative sites in later subsections.

Redundant tables (Tables 4 and 5) have been removed. As the reviewer suggested, we have corrected the mismatched metrics in the updated Figure 26 (after revision).

**RC: The innovation of this manuscript lies in local modeling, and it is recommended to provide the results of global modeling for comparative analysis to highlight the improvement effect of the method.**

AR: We acknowledge the reviewer's recognition on local fitting and the suggestion of incorporating the comparison of global vs. local fitting. To validate the improvements of our inversion framework, we have incorporated an accuracy analysis of conventional global modeling based inversion at the Howland Forest site in Figure 20, Subsection 4.2.2. The associated text at line 666 and updated figures are presented as follows:

> …To compare these results against our earlier work (Lei et al., 2019), we applied the global-fitting-based inversion method (in the left panel of Figure), which yielded an inversion accuracy of RMSE = 4.38 m at an aggregated pixel size of 0.81 ha. This demonstrates that the global-to-local two-stage inversion approach significantly improves inversion accuracy, particularly over tall forest regions…

[Figure]

**Figure 20: (a) global-fitting based inversion (Lei et al., 2019) applied to the scene ($S = 0.94, C = 9$), (b) comparing 90 m gridded maps from (a) with corresponding LVIS data; (c) interpolated 30 m gridded GEDI height map; (d) density scatterplot comparing the interpolated 30 m GEDI map with LVIS LiDAR data; (e) density scatterplot comparing ALOS-2-based inversion results with LVIS LiDAR data.**

**RC: Line 88: The wavelength of TanDEM-X is not ~0.01m, please check for updates.**

AR: Thank you for noting this typo. The wavelength is now changed to **3.1 cm** in the revised manuscript at line 90 after revision, as follows:

> …Additionally, a potential limitation of TanDEM-X observations is the insufficient penetration capability over dense forests due to the short wavelength of the X-band (~3.1 cm) (Kugler et al., 2014)…

**RC: Line 90: Are "these methods" referring to the TanDEM-X methods mentioned above? The method proposed in this paper may not necessarily outperform TanDEM-X. For example, the latest work by Qi et al (2025) adopts a strategy that is essentially similar to that of this manuscript.**

AR: We agree that our method's performance may not surpass TanDEM-X-based approaches, as the latter inherently avoid temporal decorrelation challenges in forest height estimation. However, as emphasized in the introduction, the primary contribution of focuses on large-scale and efficient Radar-LiDAR fusion for forest height mapping using open-access datasets, whereas the TanDEM-X data is not open-access at present. Additionally, L-band data offer better penetration capabilities over X-band data, enhancing the inversion accuracy for dense forest regions.

Qi et al., 2025 employs a scene-wide constant LiDAR vertical profile to enable RVOG inversion based on the approach developed by Choi et al., 2023, followed by regional model calibration to mitigate estimation

inaccuracies, a step implemented as a post-processing adjustment. Crucially, their framework does not integrate the local-scale concept into the model-based inversion itself. In contrast, our approach embeds local calibration directly into the scattering model of the inversion framework, with spatially varying model parameters dynamically determined and used for improved estimates.

This paper is the large-scale inversion application of our early developed methodologies (Lei and Siqueira, 2022; Yu et al., 2023). While the use of local-scale concepts is not new, our innovation lies in using *the GEDI local information into the physical-model based inversion framework* to enhance retrievals. This methodological integration, coupled with our focus on open-data synergy, distinguishes our approach from the post-processing regional calibration strategy in (Qi et al., 2025), both in scope and execution.

The citation to (Qi et al., 2025) and relevant discussions have been added to the revised paper at the line 87 as follows:

> …To address this, (Qi et al., 2025) proposed a post-processing correction model to refine suboptimal height estimates regionally…

References:

Qi, W., Armston, J., Choi, C., Stovall, A., Saarela, S., Pardini, M., Fatoyinbo, L., Papathanassiou, K., Pascual, A. and Dubayah, R., 2025. Mapping large-scale pantropical forest canopy height by integrating GEDI lidar and TanDEM-X InSAR data. *Remote Sensing of Environment*, *318*, p.114534.

Choi, C., Cazcarra-Bes, V., Guliaev, R., Pardini, M., Papathanassiou, K.P., Qi, W., Armston, J. and Dubayah, R.O., 2023. Large-scale forest height mapping by combining TanDEM-X and GEDI data. *IEEE Journal of Selected Topics in Applied Earth Observations and Remote Sensing*, *16*, pp.2374-2385.

Y. Lei and P. Siqueira, "Refined Forest Stand Height Inversion Approach with Spaceborne Repeat-Pass L-Band SAR Interferometry and GEDI Lidar Data," *IGARSS 2022 - 2022 IEEE International Geoscience and Remote Sensing Symposium*, Kuala Lumpur, Malaysia, 2022, pp. 6388-6391, doi: 10.1109/IGARSS46834.2022.9884755

Y. Yu, Y. Lei and P. Siqueira, "Large-Scale Forest Height Mapping in the Northeastern U.S. using L-Band Spaceborne Repeat-Pass SAR Interferometry and GEDI LiDAR Data," IGARSS 2023 - 2023 IEEE International Geoscience and Remote Sensing Symposium, Pasadena, CA, USA, 2023, pp. 1760-1763.

**RC: I don't understand why the author chose this color scheme for Figures 13 and 14, and there is a lack of corresponding explanation.**

AR: We apologize for the lack of clarity in the original maps. The earlier versions displayed forest disturbances occurring in individual years, whereas the revised version in Figure 3 and 4 now utilizes a binary map (1 = disturbance, 0 = no disturbance) to represent forest changes across the entire study period (2007–2023) with some zoomed-in close-ups in the map of northeastern China. The new figures and their captions are added as follows:

[Figure]

**Figure 3** Forest disturbance map of the New England region (2007–2023) derived from the Global Forest Change dataset (Hansen et al., 2013). The binary classification distinguishes undisturbed areas (0) from disturbed areas (1) within the period.

[Figure]

**Figure 4** Forest disturbance map of northeastern China (2007–2023) derived from the *Global Forest Change dataset* (Hansen et al., 2013). The binary classification distinguishes undisturbed areas (0) from disturbed areas (1) within the period.

**RC: Several global forest height products have been generated by combining GEDI and multi-source remote sensing data (Potapov et al, Lang et al.). The authors mentioned the limitations of these products in the introduction, and I suggest that the authors compare with these public products to highlight the performance of the proposed method and results.**

AR: We thank the reviewer for this great suggestion. We have added the comparison over the relevant performance in the Subsection 4.2.1. In specific, Table 3 summarize the accuracy evaluation metrics for the representative forest sites across the New England region, which clearly demonstrates that our product has lower bias and RMSE, thus superior to the two global GEDI-derived products.

**Table 4 Comparison of GLAD, ETH, and our ALOS-1 based canopy height products with airborne LiDAR data across all forest sites in the New England region.**

| Validation sites | | GEDI-Sentinel (ETH) | GEDI-Landsat (GLAD) | GEDI-ALOS (Our product) |
|---|---|---|---|---|
| Howland Forest | RMSE | 5.71 | 5.59 | 3.81 |
| | R2 | 0.53 | 0.1 | 0.47 |
| | Bias | 4.6 | -1.87 | -0.34 |
| | Standard Deviation | 3.38 | 5.21 | 3.78 |
| Harvard Forest | RMSE | 4.79 | 5.66 | 4.11 |
| | R2 | 0.73 | 0.24 | 0.56 |
| | Bias | 3.35 | -0.78 | 0.31 |
| | Standard Deviation | 3.45 | 4.62 | 4.10 |
| White Mountain National Forest | RMSE | 5.98 | 5.46 | 4.08 |
| | R2 | 0.58 | 0.22 | 0.33 |
| | Bias | 4.34 | -0.87 | 1.29 |
| | Standard Deviation | 4.04 | 5.2 | 3.87 |
| Green Mountain National Forest | RMSE | 5.55 | 5.78 | 4.97 |
| | R2 | 0.69 | 0.18 | 0.53 |
| | Bias | 3.93 | -1.79 | 0.46 |
| | Standard Deviation | 3.92 | 5.09 | 4.95 |
| Naugatuck State Forest | RMSE | 4.89 | 5.07 | 5.06 |
| | R2 | 0.67 | 0.45 | 0.47 |
| | Bias | 2.83 | 1.16 | -0.84 |
| | Standard Deviation | 3.98 | 4.43 | 5.01 |

**RC: There are several typos in the current manuscript. For example, Line 235: This finding i based on…; Line 445 Table3: left column, etc. Please proofread the manuscript carefully.**

AR: Thanks. We have fixed several typos after proofreading carefully.